# Oligodendrocytes support axonal transport and maintenance via exosome secretion

**Carsten Frühbeis**[1◉], **Wen Ping Kuo-Elsner**[1,2◉], **Christina Müller**[1], **Kerstin Barth**[1], **Leticia Peris**[3], **Stefan Tenzer**[4], **Wiebke Möbius**[5], **Hauke B. Werner**[5], **Klaus-Armin Nave**[5], **Dominik Fröhlich**[1,6], **Eva-Maria Krämer-Albers**[1,2]*

1 Institute of Developmental Biology and Neurobiology, Johannes Gutenberg University of Mainz, Mainz, Germany, 2 Focus Program Translational Neuroscience, Johannes Gutenberg University of Mainz, Mainz, Germany, 3 Grenoble Institut des Neurosciences, Université Grenoble Alpes, Inserm, U1216, Grenoble, France, 4 Institute of Immunology, University Medical Center, Johannes Gutenberg University of Mainz, Mainz, Germany, 5 Department of Neurogenetics, Max Planck Institute of Experimental Medicine, Göttingen, Germany, 6 Translational Neuroscience Facility and Department of Physiology, School of Medical Sciences, University of New South Wales, Sydney, New South Wales, Australia

◉ These authors contributed equally to this work.
* alberse@uni-mainz.de

**Data Availability Statement:** All relevant data are within the paper and its Supporting Information files.

## Abstract

Neurons extend long axons that require maintenance and are susceptible to degeneration. Long-term integrity of axons depends on intrinsic mechanisms including axonal transport and extrinsic support from adjacent glial cells. The mechanisms of support provided by myelinating oligodendrocytes to underlying axons are only partly understood. Oligodendrocytes release extracellular vesicles (EVs) with properties of exosomes, which upon delivery to neurons improve neuronal viability in vitro. Here, we show that oligodendroglial exosome secretion is impaired in 2 mouse mutants exhibiting secondary axonal degeneration due to oligodendrocyte-specific gene defects. Wild-type oligodendroglial exosomes support neurons by improving the metabolic state and promoting axonal transport in nutrient-deprived neurons. Mutant oligodendrocytes release fewer exosomes, which share a common signature of underrepresented proteins. Notably, mutant exosomes lack the ability to support nutrient-deprived neurons and to promote axonal transport. Together, these findings indicate that glia-to-neuron exosome transfer promotes neuronal long-term maintenance by facilitating axonal transport, providing a novel mechanistic link between myelin diseases and secondary loss of axonal integrity.

## Introduction

Extracellular vesicles (EVs) are a heterogeneous group of secreted vesicles that appear to be engaged in a wide range of neural cell communication processes and have been implicated in neural development, maintenance, neurodegeneration, and regeneration [1–3]. According to their site of biogenesis, EVs can be subclassified as exosomes, which are released from the lumen of secretory multivesicular bodies (MVBs), and microvesicles, which are shed from the plasma membrane [4,5]. By delivering a mixed cargo of biomolecules including nucleic acids

**Funding:** This work was supported by the Deutsche Forschungsgemeinschaft (grants KR 3668/1-1, KR 3668/1-2; https://www.dfg.de/) to EMKA. CF received intramural funding from the Johannes Gutenberg University Mainz (https://www.uni-mainz.de/). WPKE was supported by a PhD-fellowship from the Focus Program Translational Neuroscience, Johannes Gutenberg University Mainz (https://www.blogs.uni-mainz.de/ftn-eng/). CM was supported by a postdoctoral fellowship from the Carl Zeiss Stiftung (https://www.carl-zeiss-stiftung.de/english/index.html). KAN acknowledges research support from the European Research Council (Advanced Grant MyeliNANO; https://erc.europa.eu/) and Adelson Medical Research Foundation (http://www.adelsonfoundation.org/nrrfc.html). The funders had no role in study design, data collection and analysis, decision to publish, or preparation of the manuscript.

**Competing interests:** The authors have declared that no competing interests exist.

**Abbreviations:** 10K, 10,000*g*-centrifuged; 100K, 100,000*g*-centrifuged; AAV, adeno-associated virus; CNP, 2′,3′-cyclic nucleotide 3′-phosphodiesterase; CNS, central nervous system; DIV, day in vitro; E, embryonic day; EV, extracellular vesicle; LC, liquid chromatography; MS, mass spectrometry; MTT, 3-(4,5-dimethylthiazol-2-yl)-2,5-diphenyltetrazolium bromide; MVB, multivesicular body; NTA, nanoparticle tracking analysis; PLP, proteolipid protein; sEV, small extracellular vesicle.

to target cells, EVs are able to elicit pleiotropic effects and give rise to plastic modulation of the tissue microenvironment [6]. In the central nervous system (CNS), neurons and glial cells release EVs under physiological and pathological conditions. However, the target cell responses and the mode of action of EVs in the CNS are not well understood.

Our previous work demonstrated the transfer of exosomes from myelinating oligodendrocytes to neurons [7,8]. Triggered by neuronal signals, oligodendrocytes release small EVs (sEVs) with features of exosomes derived from MVBs. These MVBs appear prevalent at periaxonal sites in myelinated nerves. Neurons internalize oligodendroglial sEVs by endocytosis (which can occur at somatodendritic and axonal sites) and recover the EV cargo, thereby importing bioactive molecules. Notably, the metabolic activity of neurons receiving sEVs is increased, in particular under stress conditions [8]. In addition, the neuronal firing rate is increased, and pro-survival signaling pathways are activated [9]. Together, these observations suggest that glia-derived sEVs of exosomal origin mediate neuroprotection and improve neuronal homeostasis.

Neurons exhibit a complex polarized architecture, with axons projecting far away from the neuronal cell body that need to be supplied with energy and newly synthesized molecules. The long-term integrity of myelinated axons depends on oligodendroglial factors, which has been generally described as glial trophic support [10]. Mice lacking genes encoding oligodendroglial proteolipid protein (PLP) and 2′,3′-cyclic nucleotide 3′-phosphodiesterase (CNP) develop a secondary progressive axonal degeneration characterized by the formation of axonal swellings [11,12], which may result from a defect in axonal transport [13]. Glial support has been suggested to reflect the supply of axons with energy-rich glycolytic metabolites such as lactate and pyruvate through the monocarboxylate transporters MCT1 and MCT2 [14–17]. However, the causal relationship between the lack of PLP and CNP, metabolic support, and axonal degeneration is still unknown.

Axonal transport drives cargo along microtubules in both the anterograde and retrograde direction and is essential for neuronal homeostasis. It comprises fast axonal transport of vesicular cargo and a slow component for the transport of cytoskeletal components and soluble content [18,19]. Reduced transport rates are an early sign of axonal dysfunction preceding axonal degeneration and are also an early event in human myelin diseases [20–24]. As yet, the molecular mechanisms linking glial dysfunction and axonal transport decline are unknown.

Here, we studied the effect of oligodendroglial sEVs on axonal maintenance and examined their impact on fast axonal transport by live cell imaging of vesicular cargo traveling along axons. We show that adding sEVs to neurons promotes axonal transport most prominently under conditions of cell stress such as nutrient deprivation. Intriguingly, we found that the release of sEVs from PLP- and CNP-deficient oligodendrocytes is impaired and that mutant sEVs, which share a common altered proteome signature, are dysfunctional in terms of neuronal support. We conclude that sEVs of exosomal origin delivered from oligodendrocytes to neurons contribute to axonal homeostasis and long-term maintenance by facilitating axonal transport. Secondary axonal degeneration, as observed in PLP- and CNP-deficient mice, is connected to impaired glial exosome function, providing a so far missing link between glial dysfunction, axonal transport, and axonal degeneration as seen in various neurological diseases.

## Results

### Oligodendroglial sEVs with exosomal properties promote fast axonal transport

To explore the functional role of oligodendroglial exosomes in oligodendrocyte–neuron interaction and glial support, we examined their impact on fast axonal transport. We performed

live imaging in developing primary hippocampal neurons cultured at low density and determined the dynamics of vesicles delivering brain-derived neurotrophic factor (BDNF), exemplifying dense core vesicles [25,26]. sEVs were isolated from culture supernatants of primary oligodendrocytes by differential ultracentrifugation (Fig 1A and 1B). As EV isolation procedures do not selectively purify exosomes, we refer to sEVs when studying these fractions [27], although our work indicates that these sEVs largely comprise exosomes derived from MVBs (Fig 1 and see [7,8,28]). Characterization of sEVs isolated via differential ultracentrifugation (100,000$g$-centrifuged [100K] pellet), immuno-bead isolation from 10,000$g$-centrifuged (10K) supernatants, sucrose density gradient centrifugation, and size exclusion chromatography demonstrates that the purified sEV fractions carry the relevant sEV markers CD81, CD9, and PLP, while markers that would indicate non-exosomal co-isolating particles are absent (Fig 1B–1E). Transmission electron microscopy of the oligodendrocyte extracellular milieu reveals homogeneous groups of small vesicles consistent with exosomes in the extracellular space (Fig 1F).

In a set of pilot experiments, developing primary hippocampal neurons (day in vitro [DIV] 2–3) expressing BDNF-mCherry were exposed to oligodendrocyte-derived sEVs isolated by ultracentrifugation and recorded after short-term incubation by time-lapse imaging to determine the immediate effect of sEVs on the movement of BDNF-mCherry-carrying transport vesicles along axons (S1 Fig). Axons were identified according to Dotti et al. [29], and imaging was performed at stage 3 of axon determination. Quantitative analysis of single vesicle trajectories in kymographs revealed a similar distribution profile of vesicular movement and vesicle velocities in sEV treatment and untreated conditions. However, sEV treatment appeared to decrease the pausing time, i.e., the percentage of time pauses were exhibited by every single vesicle during its total trajectory (S1E Fig). These data indicate that oligodendroglial sEVs can influence axonal transport by reducing vesicle pausing time. In a second trial, we studied axonal transport under stress conditions by challenging sEV-receiving neurons with oxidative stress, which is commonly associated with neurodegeneration [30]. Our previous results had shown that preconditioning neurons with oligodendroglial sEVs protected neurons from harm [8]. Applying the same sEV treatment protocol to BDNF-mCherry-expressing hippocampal neurons (15 h of pretreatment followed by 1 h of oxidative stress and imaging), we found that oligodendroglial sEVs significantly increased anterograde vesicle movement and decreased the pool of static vesicles, while retrograde movement and velocities were unaffected (S2 Fig). The relative pausing time of transport vesicles appeared reduced in neurons treated with oligodendroglial sEVs, but this reduction did not reach statistical significance. In a control setting, HEK-derived sEVs did not exhibit similar effects, indicating a cell-type-specific activity on axonal transport of oligodendroglial sEVs (S2 Fig). These results suggest that oligodendroglial sEVs promote fast axonal transport by reducing vesicle pauses and enhancing anterograde vesicle movement, most prominently when neurons are subjected to stress. Neurons maintained under full medium conditions are provided with excess of nutrients and growth factors, and therefore it may be hard to detect additional factors promoting axonal transport under full medium conditions.

## Oligodendroglial sEVs maintain axonal transport in starving neurons

Based on these initial findings, we decided to study the impact of oligodendroglial sEVs on axonal transport under conditions of nutrient deprivation (by depletion of the medium supplement B27), reflecting the withdrawal of neurons from the external supply of growth-promoting factors essential for their maintenance. To avoid sEV pelleting and compression, we employed 10K cell culture supernatants as a source of sEVs in a native state, as originally

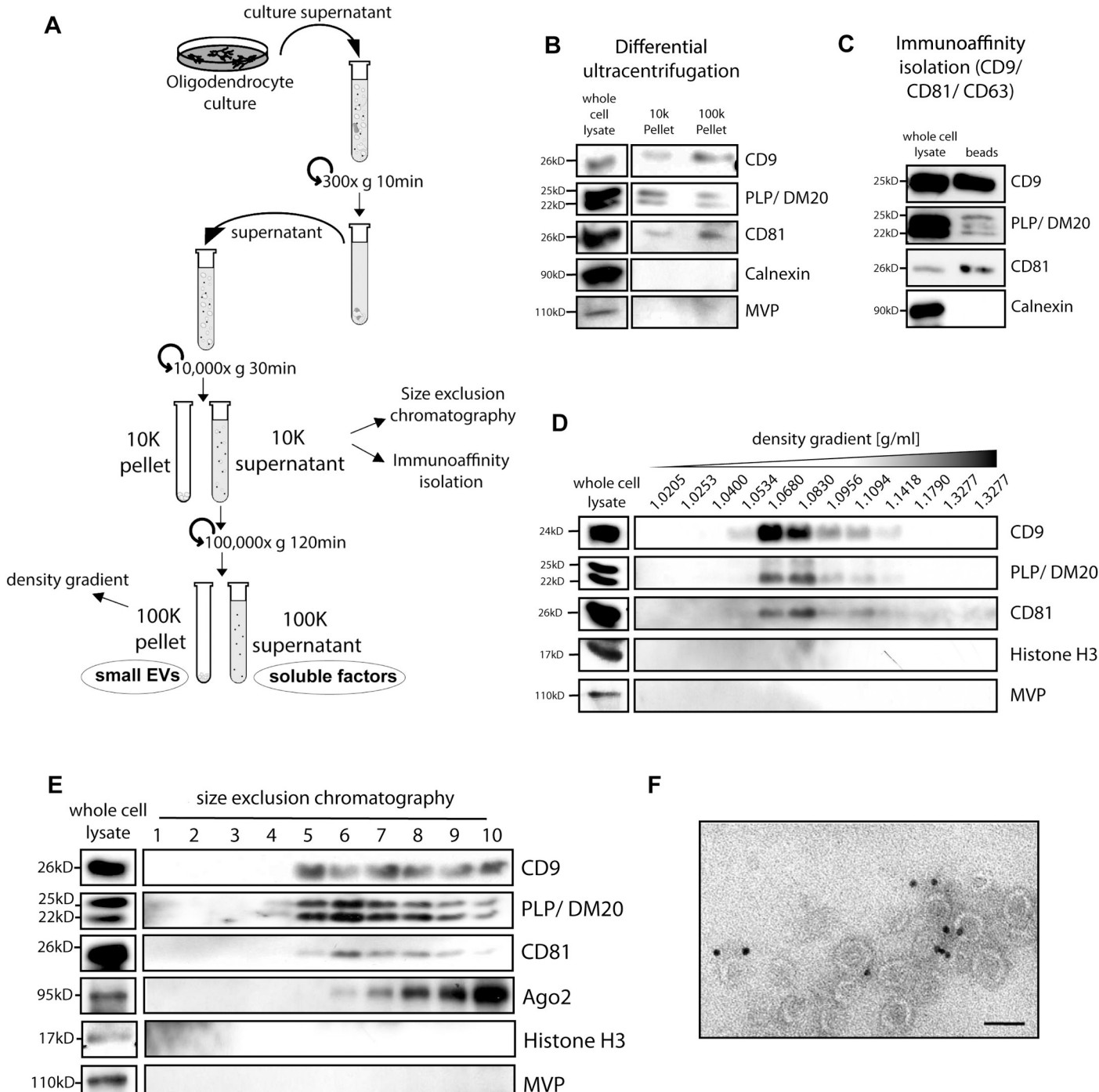

**Fig 1. Characterization of oligodendrocyte-derived sEVs.** (A) Schematic illustration of differential ultracentrifugation and the generation of 10K and 100K supernatants and pellets. (B–E) Western blot analysis of sEVs isolated by differential ultracentrifugation (B), immuno-bead isolation from 10K supernatants (C), density gradient centrifugation of 100K pellets (D), or size exclusion chromatography of 10K supernatant (E). Underlying images of blots can be found in S1 Images. (F) Immuno-transmission electron microscopy micrograph depicting MAG-labeled sEVs characteristic of exosomes in the extracellular space of cultured oligodendrocytes (MAG is an oligodendroglia-specific membrane protein present in exosomes). Bar, 100 nm. 10K, 10,000*g*-centrifuged; 100K, 100,000*g*-centrifuged; EV, extracellular vesicle; MAG, myelin-associated glycoprotein; PLP, proteolipid protein; sEV, small extracellular vesicle.

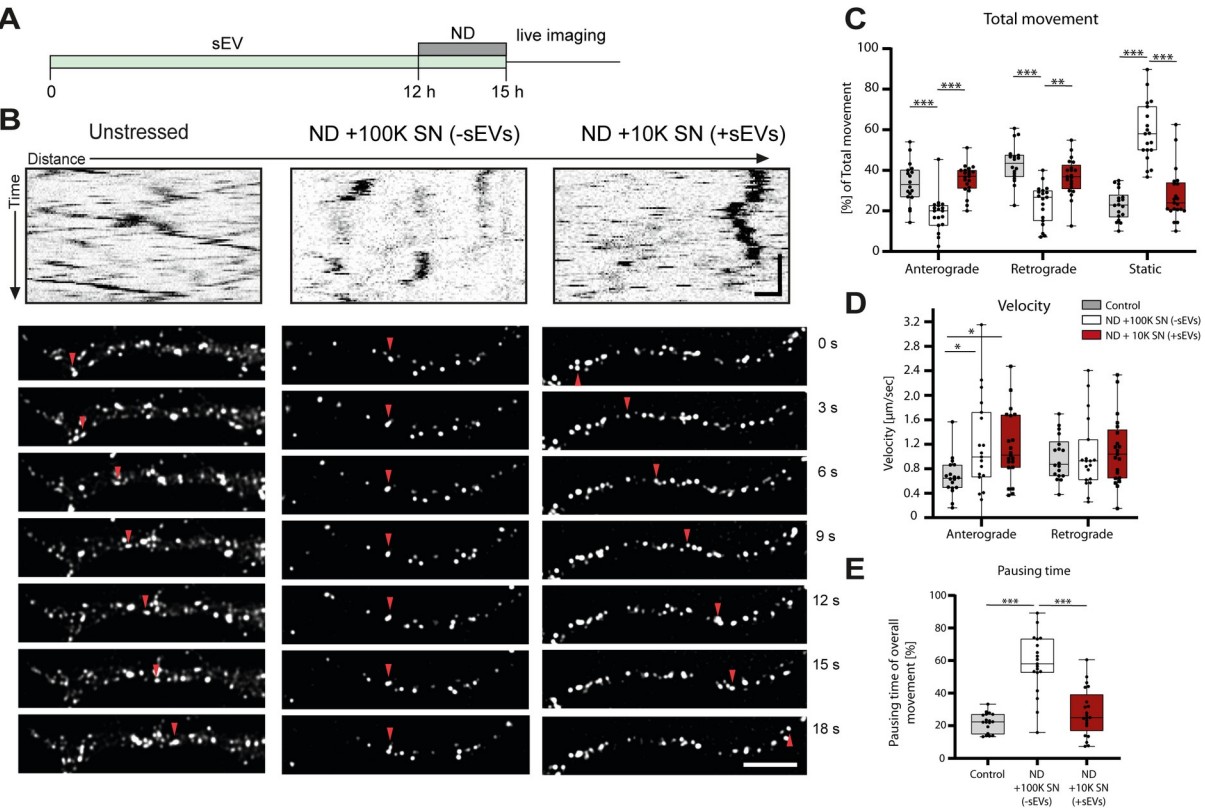

**Fig 2. Influence of sEVs on axonal transport under starvation stress.** (A) Illustration of experimental schedule. Hippocampal neurons were treated with sEVs and subjected to starvation stress by depletion of B27 supplement (nutrient deprivation) during the last 3 h before live imaging. All treatment conditions were identical regarding the presence of oligodendrocyte-derived soluble factors and differed in presence of sEVs. (B) Representative kymographs and corresponding time-lapse frames illustrating movement of BDNF-mCherry-positive vesicles generated from hippocampal neurons exposed to nutrient deprivation. Neurons were kept in complete medium (unstressed) or subjected to nutrient deprivation in the absence [ND + 100K SN (−sEVs)] or presence of sEVs [ND + 10K SN (+sEVs)]. Red arrowhead follows 1 distinct particle moving over time. Kymograph: horizontal scale bar = 5 μm, vertical scale bar = 10 s; fluorescent picture frames: scale bar = 5 μm (C–E). Quantitative analysis of kymographs regarding total movement (C), velocity (D), and pausing time (E) of BDNF-mCherry-positive vesicles. Data are presented as median, boxes (25th percentile to 75th percentile), and whiskers (minimum to maximum showing all data points), $n = 18$ recorded neurons for control, $n = 19$ for ND + 100K SN (−sEVs), $n = 21$ for ND + 10K SN (+sEVs), from 3 independent experiments. $^*p < 0.05$, $^{**}p < 0.01$, $^{***}p < 0.001$, Shapiro–Wilk normality test following Kruskal–Wallis test with Dunn's multiple comparison test. Underlying data can be found in S1 Data. 10K, 10,000$g$-centrifuged; 100K, 100,000$g$-centrifuged; ND, nutrient deprivation; sEV, small extracellular vesicle; SN, supernatant.

released by the cells. These 10K supernatants are largely devoid of cellular debris and larger particles including organelles, apoptotic bodies, and large plasma-membrane-derived extracellular vesicles (Fig 1A). As a control, we used 100K supernatants, where sEVs are depleted by ultracentrifugation but would contain oligodendrocyte-derived soluble factors that may have an influence in the assay [31]. Primary hippocampal neurons were treated with sEV-conditioned oligodendrocyte supernatant (10K +sEVs) or sEV-deprived supernatant (100K −sEVs) and subjected to nutrient deprivation (ND) during the last 3 h of the treatment before time-lapse imaging (Fig 2). Untreated neurons grown in full culture medium served as reference for optimal conditions. As expected, nutrient deprivation resulted in a significant decline of overall vesicular movement (Fig 2B–2E, white boxes). Intriguingly, axonal transport was almost completely restored in nutrient-deprived neurons in the presence of oligodendroglial sEVs (Fig 2B–2E, red boxes). We observed a significant increase in anterograde as well as retrograde movement, and the number of static vesicles was substantially reduced upon sEV treatment,

back to the normal unstarved level (Fig 2C). Vesicle velocity was not affected by sEV treatment, although it seemed that the 100K control supernatant as well as the 10K treatment supernatant appeared to increase anterograde vesicle velocity, which may be attributed to the influence of oligodendrocyte-derived soluble factors present in both supernatants (Fig 2D). Furthermore, the presence of sEVs significantly decreased the pausing periods of vesicles in starving neurons (Fig 3E). Thus, neurons receiving oligodendroglial sEVs are able to maintain axonal transport even under conditions of deprivation of external nutrients.

## Exosome secretion is impaired in mice affected by secondary axonal degeneration

We further made use of PLP- and CNP-deficient mice, which are established mouse models of glia-dependent secondary axonal degeneration [32]. It is still unclear how the lack of oligodendroglial PLP and CNP is related to the degeneration of axons. Since PLP and CNP are both sorted into exosomes [7], we investigated exosome release in the null mice. We first looked in the myelinated tract in situ at the appearance of MVBs, resembling the cytoplasmic storage sites of exosomes before their actual release. Oligodendroglial MVBs can be readily detected in ultrathin sections of optic nerve fibers by electron microscopy, although their detection depends on their presence in the plane of the section (Fig 3A). Quantification of MVBs in optic nerve sections revealed a 3.6-fold higher number of oligodendroglial MVBs in PLP-null as compared to wild-type mice (Fig 3B), while the number of MVBs was unchanged in CNP-null mice (note that MVBs will be strongly underrepresented in relation to axons in the sections). In wild-type optic nerves, MVBs are prevalent at periaxonal (adaxonal) sites, which are the site of release for axonal delivery. However, in CNP-null mice MVBs appeared underrepresented in the adaxonal and abaxonal cytoplasmic domains, while they were more frequently detected in cytoplasmic channels within myelin (myelinic channels) and in the cell bodies (Fig 3C). Thus, trafficking of MVBs to periaxonal sites appears impeded in CNP-null mice, indicating that fewer MVBs are available at these distant sites for exosome release. On the other hand, MVBs in PLP-null mice were more frequent in myelinic channels, at abaxonal loops, and in cell bodies, indicating a general accumulation of MVBs in oligodendrocytes (Fig 3D). The overall ultrastructure and appearance of MVBs was similar between wild-type and mutant mice. These observations demonstrate that oligodendroglial MVBs are of normal morphology, but are mislocalized in myelinated tracts of PLP-null as well as CNP-null mice, suggesting a defect in exosome release.

We then examined exosome release in cultured oligodendrocytes derived from PLP- and CNP-null mice. Mutant oligodendrocytes exhibited normal morphological differentiation in culture, expressed the expected lineage markers, and appeared equally viable, although cell death (as determined by lactate dehydrogenase leakiness into the culture medium) appeared slightly increased in PLP-null oligodendrocytes (S3 Fig). We isolated sEVs including exosomes from culture supernatants collected from wild-type, PLP-null, and CNP-null oligodendrocytes by differential centrifugation and determined the particle number and size by nanoparticle tracking analysis (NTA). The number of released particles was significantly reduced in PLP- and CNP-null oligodendrocytes, while the size distribution was unchanged (Fig 3E). Over 24 h, wild-type, CNP-null, and PLP-null oligodendrocytes secreted 435 (± 26), 262 (± 43), and 214 (± 22) particles per cell, respectively, in the size range of 50 to 150 nm, indicating approximately a 50% reduction of sEV release by mutant oligodendrocytes. Western blot analysis of isolated sEVs normalized to the number of releasing cells revealed that common EV markers including Alix, Tsg101, Flotillin-1 (Flot-1), Hsp70, PLP, CNP, and Sirtuin-2 (Sirt2) were decreased in mutant compared to wild-type sEVs, confirming the decreased release rate (Fig

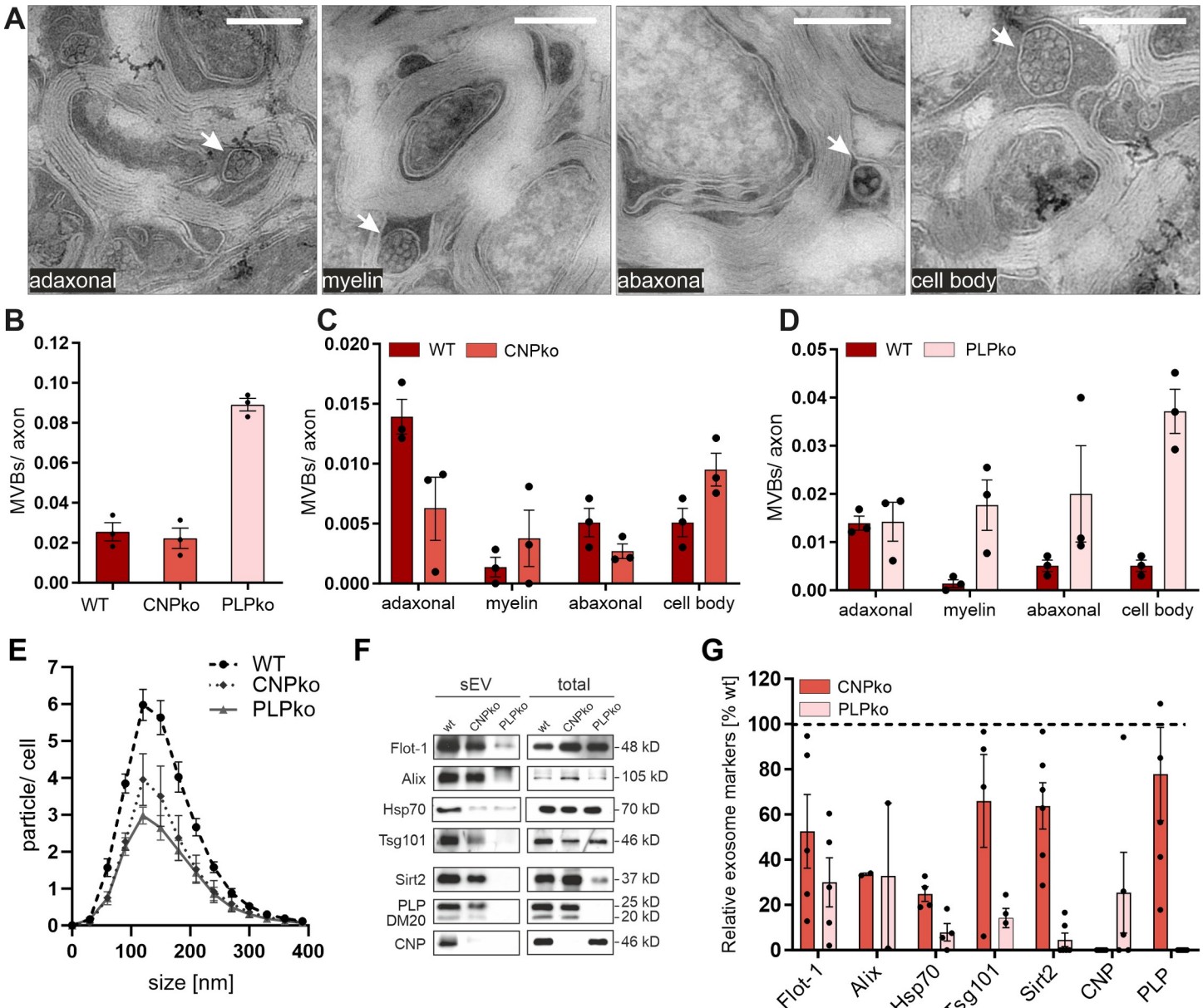

**Fig 3. Impaired exosome release from PLP- and CNP-null oligodendrocytes.** (A) Representative electron microscopy images taken from optic nerve cross-sections demonstrating MVBs (arrows) in different oligodendrocyte compartments (from left to right): in the inner cytoplasmic loop close to the axon (adaxonal), in cytoplasmic channels within myelin (myelin), in the outer cytoplasmic loop (abaxonal), or in oligodendrocyte cell bodies (scale bar 500 nm). (B–D) Quantification of oligodendrocyte MVBs counted in optic nerves of wild-type (WT), CNP-null (CNP knockout [CNPko]), and PLP-null (PLP knockout [PLPko]) animals and depicted as (B) overall number of MVBs per axon or (C and D) relative number of MVBs located in different compartments as indicated in (A). SEM, $n = 3$. (E) Nanoparticle tracking analysis of sEVs isolated from WT, CNP-null, and PLP-null oligodendrocytes normalized to the number of secreting cells. SEM, $n = 3$. (F) Representative Western blot of isolated sEVs derived from equal numbers of WT, CNP-null, and PLP-null oligodendrocytes and (G) densitometric quantification. Signals from sEVs were normalized to the total cell lysate of corresponding sEV-secreting cells and expressed in relation to wild-type levels (dashed line). SEM, $n = 2$–5. Underlying data and images of blots can be found in S1 Data and S1 Images, respectively. CNP, 2′,3′-cyclic nucleotide 3′-phosphodiesterase; MVB, multivesicular body; PLP, proteolipid protein; SEM, standard error of the mean; sEV, small extracellular vesicle; WT, wild-type.

3F and 3G). The density of sEVs released by mutant oligodendrocytes appeared normal, as demonstrated by density gradient analysis, indicating that the lipid/protein ratio and sEV sub-types are similar between wild-type and mutant sEVs (S4 Fig). In summary, these data indicate that PLP- and CNP-null oligodendrocytes secrete considerably lower levels of sEVs/exosomes.

## CNP- and PLP-null sEVs share common proteome abnormalities

To investigate potential differences in cargo composition between wild-type and mutant sEVs, we performed a differential proteome analysis of density-gradient-purified sEVs by label-free quantitative liquid chromatography (LC)–mass spectrometry (MS). Gradient-purified sEVs should be enriched for exosomes and devoid of non-EV components co-sedimenting in 100K pellets. While gradient-isolated sEVs unfortunately lose their functional activity during processing (most likely due to osmotic effects), impeding their application in axonal transport assays, this additional step of sEV purification is thought to improve the quality of proteomics results. The sEV input into LC-MS was normalized according to the total number of sEV-secreting cells (sEV pools isolated from $2.3 \times 10^8$ cells), which originated from independent cell preparations and several mouse brains. Functional annotation to Gene Ontology (GO) parent terms "cellular components" and "biological processes" assigned the largest fraction of identified proteins to the categories "extracellular exosome" and "transport" (reflecting the ability of exosomes to deliver cargo between cells), respectively, indicating that the isolation procedure was efficient (Fig 4A).

To reveal alterations in the proteome of sEVs derived from mutant PLP- and CNP-null oligodendrocytes, signal intensities were normalized and expressed in parts per million of total protein (ppm). Differential analysis revealed that a number of proteins were up- or downregulated in mutant versus wild-type sEVs (Fig 4B–4D; S2 Data; S5 Fig). STRING analysis demonstrates highly interconnected networks among up- and downregulated proteins in CNP-null as well as PLP-null sEVs, highlighting the tight relationship between these proteins, most likely linked to potentially affected biological pathways (S6–S9 Figs). Interestingly, regulated proteins overlapped to a high degree: 50 out of 80 significantly upregulated proteins and 6 out of 25 downregulated proteins were common between PLP-null and CNP-null sEVs/exosomes (Fig 4C). The downregulated proteins include Annexin A5 (ANXA5), Heat shock-related 70 kD protein 2 (Hsp70-2), Sirtuin-2 (Sirt2), Immunoglobulin superfamily member 8 (IGSF8), and Peptidyl-prolyl cis-trans isomerase (FKB1A). Notably, these 5 candidates belong to the top-10 downregulated proteins in both mutant sEV types and were downregulated at least 2- to 5-fold (Fig 4D). Consistent with normalization to total sEV input, typical sEV markers such as the tetraspanins CD81 and CD63 were detected in similar levels in wild-type and mutant sEVs (Alix and Tsg101 were not identified by LC-MS).

We further focused on downregulated candidates and attempted to validate the proteins found most prominently reduced in CNP- and PLP-null sEVs by semiquantitative Western blotting. sEVs isolated by differential ultracentrifugation were normalized to particle number by NTA before being subjected to Western blot analysis (Fig 4E). In addition to Hsp70-2 (antibody recognizing Hsp70 family including Hsp70-2), ANXA5, IGSF8, and Sirt2, we examined the sEV markers Flotillin-1 (Flot-1), Alix, and Tsg101 that were not revealed by LC-MS. Detection levels of PLP were equal between wild-type and CNP-null sEVs, and, furthermore, levels of CNP were equal between wild-type and PLP-null sEVs, confirming equivalent input of sEV particles. Flot-1, Tsg101, ANXA5, and Sirt2 appeared reduced in PLP-deficient sEVs (Fig 4F). Alix, Hsp70, and IGSF8 were reduced in both PLP- and CNP-null sEVs. While these data largely confirm the results obtained by proteomics, it should be noted that the semiquantitative Western blot data exhibit high variance. Together, null-mutant-derived sEVs are impaired not only in quantitative but also in qualitative terms. PLP- and CNP-null sEVs exhibit an altered protein composition and, furthermore, share a remarkably common signature.

## Cargo delivery by PLP- and CNP-null sEVs

We further asked whether mutant-derived sEVs are malfunctioning and fail to deliver signals to neurons. First, we addressed whether PLP- and CNP-null sEVs are able to deliver their

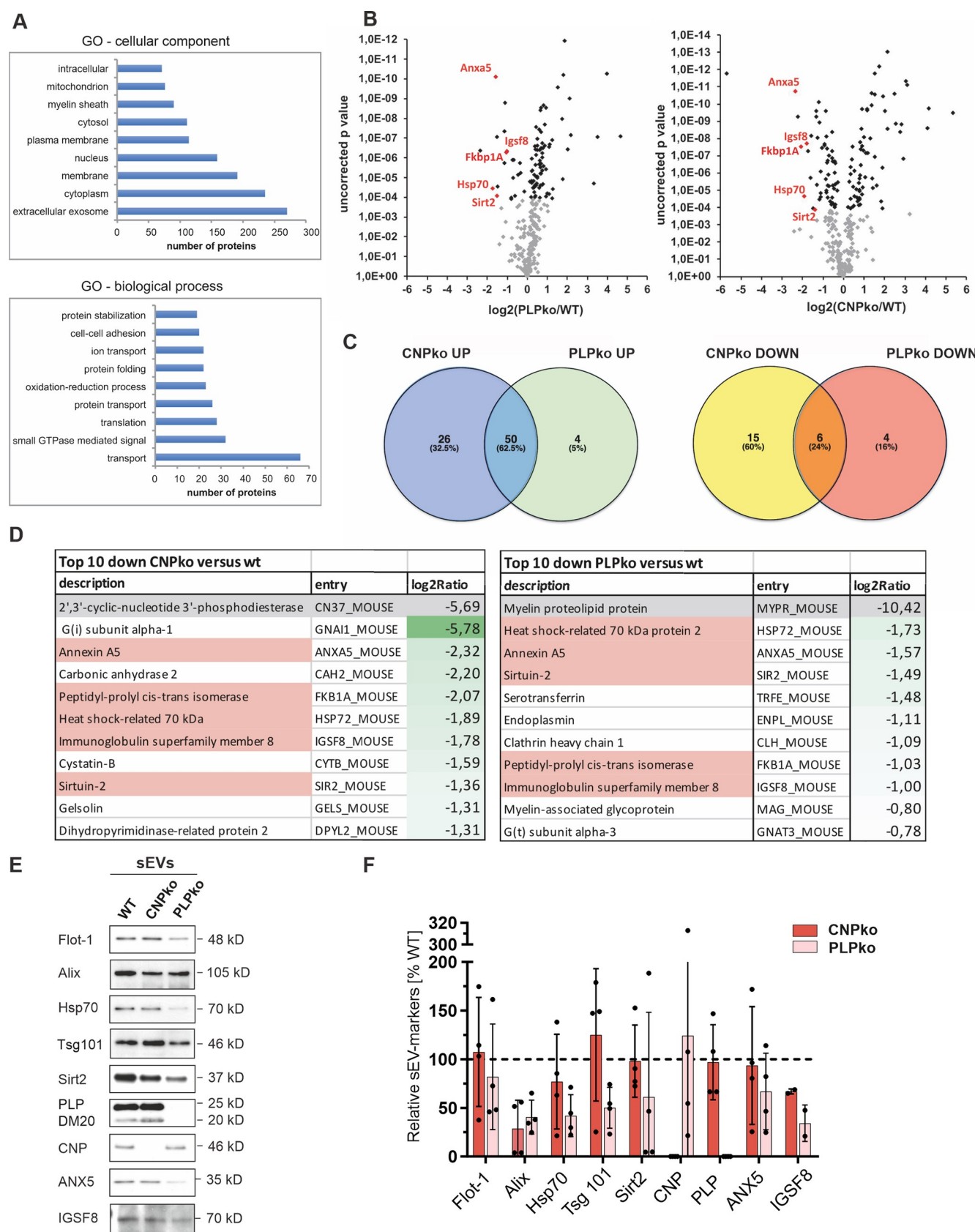

**Fig 4. Differential proteome analysis of wild-type PLP- and CNP-null exosomes.** LC-MS analysis of density-gradient-purified sEVs. (A) Functional annotation to Gene Ontology terms "cellular component" and "biological process". (B) Volcano blot of proteins over- or underrepresented in null-mutant (PLP knockout [PLPko] and CNP knockout [CNPko]) versus wild-type (WT) sEVs. Proteins downregulated in both PLPko and CNPko sEVs are labeled in red. (C) Venn diagram illustrating overlap of up- and downregulated proteins between PLPko and CNPko sEVs. (D) List of top-10 downregulated proteins in PLPko and CNPko sEVs and their level of downregulation expressed as log2 fold change (reflected by green color intensity). PLP and CNP are at the top of lists due to genetic deletion (highlighted in grey). Proteins underrepresented in both PLPko and CNPko sEVs are highlighted in red. Underlying proteomics data can be found in S2 Data. (E) Western blot analysis of sEVs isolated from WT, CNPko, and PLPko oligodendrocytes normalized to particle number and (F) densitometric quantification (*n* = 2–4). Protein levels are expressed relative to wild-type (dashed line). Underlying images of blots and data can be found in S1 Images and S1 Data, respectively. CNP, 2′,3′-cyclic nucleotide 3′-phosphodiesterase; GO, Gene Ontology; LC, liquid chromatography; MS, mass spectrometry; PLP, proteolipid protein; sEV, small extracellular vesicle; WT, wild-type.

cargo. To this end, we utilized Cre as a reporter of sEV transfer and cargo retrieval in neurons, as established previously [8]. Adeno-associated virus (AAV)–transduced primary oligodendrocytes expressing Cre-recombinase were co-cultured in Boyden chambers with primary cortical neurons that carry GFP as reporter, which is only expressed after Cre-mediated recombination. Delivery of Cre via oligodendroglial sEVs and recovery in neurons will thus activate the GFP reporter. This assay is thus a measure of the efficiency of sEVs to deliver functional cargo to target neurons. Indeed, we observed reporter expression in neurons co-cultured with wild-type as well as PLP- and CNP-null oligodendrocytes, indicating that mutant sEVs were internalized by neurons and released Cre (Fig 5A). However, reporter expression in target neurons was reduced by approximately half upon co-culture with CNP-null oligodendrocytes and PLP-null oligodendrocytes (Fig 5B). Thus, sEVs/exosomes released by PLP- and CNP-deficient oligodendrocytes are competent to deliver cargo to neurons, although the level of transfer is decreased, most likely reflecting the reduced rate of secretion by the mutant donor cells.

## PLP- and CNP-null sEVs lack the ability to support nutrient-deprived neurons

Previously, we have shown that oligodendroglial exosomes/sEVs are neuroprotective and mediate stress resistance. Neurons exposed to starvation maintain higher levels of metabolic activity when treated with sEVs during the nutrient-deprived period [8]. To examine if sEV-mediated support of nutrient-deprived neurons is dose-dependent, we generated a dilution-series of sEV-containing (10K) supernatants collected from wild-type oligodendrocytes and measured their effect on the viability of primary cortical neurons by 3-(4,5-dimethylthiazol-2-yl)-2,5-diphenyltetrazolium bromide (MTT) assay (Fig 5C). Reduction of sEVs to 60% of initial particle concentration ($100\% = 2.48 \times 10^8 \pm 0.12 \times 10^8$ particles/ml) was sufficient to largely abrogate the sEV-mediated support, comparably to sEV-depleted 100K supernatants (0%). Since sEV release in CNP- and PLP-null mice is reduced by roughly half, we hypothesized that the reduced level of sEV transfer from mutant oligodendrocytes would result in loss of support of nutrient-deprived neurons. We treated cortical neurons during starvation with sEV-containing 10K supernatants derived from equal numbers of wild-type, PLP-null, and CNP-null oligodendrocytes. As described above, the mutant-derived supernatants would contain roughly halved sEV concentration. Analysis of cell viability revealed that wild-type sEV-containing supernatants were able to fully restore the metabolic activity of starving cells, while PLP- and CNP-null sEV-containing supernatants lacked the ability to support nutrient-deprived neurons, as do wild-type-derived supernatants that were depleted of sEVs (Fig 5D).

To examine if the observed qualitative differences between wild-type and mutant sEVs could also contribute to the lack of neuronal support, starving cortical neurons were treated with sEV-containing (10K) supernatants that were normalized for particle concentration by

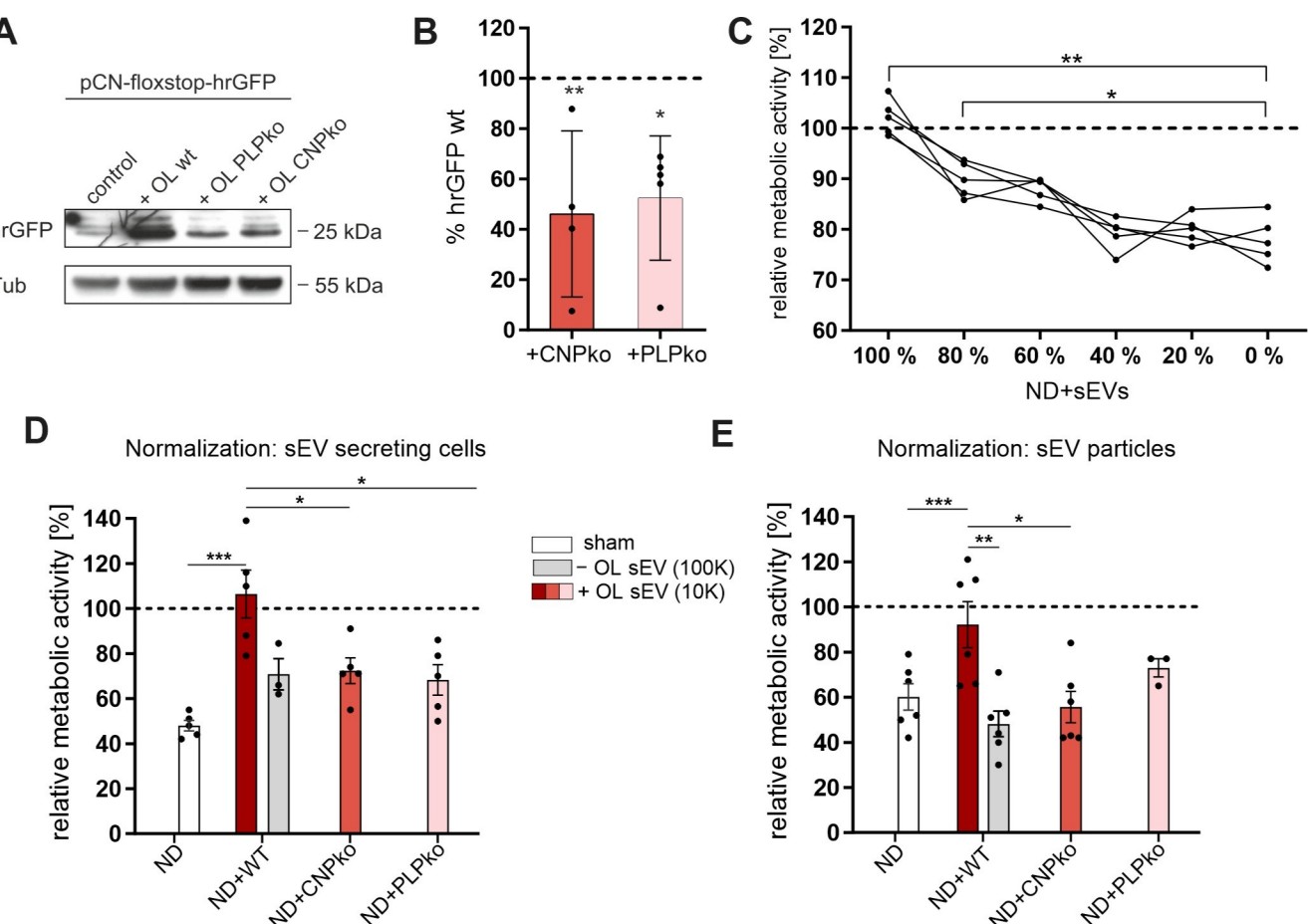

**Fig 5. PLP- and CNP-null exosomes deliver cargo but lack functional activity.** (A) Western blot showing hrGFP reporter gene activation in primary cortical neurons upon exposure to Cre-labeled oligodendroglial (OL) sEVs secreted from equal numbers of wild-type (WT), CNP-null (CNP knockout [CNPko]), or PLP-null (PLP knockout [PLPko]) oligodendrocytes. Control shows background in neurons not receiving Cre sEVs. Alpha-tubulin (Tub), loading control. (B) Quantitative representation of hrGFP Western blot signals normalized to alpha-tubulin and expressed relative to wild-type signal (dashed line). SEM, $n = 4$–5, ANOVA followed by Dunn's multiple comparison test, $^*p < 0.05$, $^{**}p < 0.001$. (C) Dose–response curve of sEV-mediated protection of cortical neurons from nutrient deprivation (ND). Metabolic activity was measured in neurons receiving a series of dilutions of sEV-containing supernatants derived from wild-type oligodendrocytes (100% = undiluted 10K supernatants, 0% = sEV-deprived 100K supernatants). Dashed line, reference of unstarved neurons ($n = 5$, SEM), 1-way ANOVA test with Dunn's multiple comparison test, $^*p < 0.05$, $^{**}p < 0.001$. (D and E) Metabolic activity of cortical neurons subjected to nutrient deprivation (ND, sham) treated with sEV-containing 10K supernatants (+OL sEV) derived from wild-type (WT), CNP-null (CNPko), or PLP-null (PLPko) oligodendrocytes. Grey bar compares treatment with sEV-depleted 100K supernatants (−OL sEV) derived from wild-type (WT) oligodendrocytes. Dashed line, reference of unstarved neurons. sEV input was normalized according to (D) the number of sEV-secreting cells ($n = 3$–5, SEM) or (E) the number of particles applied to neurons, as determined by NTA ($n = 3$–6), 2-tailed Student $t$ test, $^*p < 0.05$, $^{**}p < 0.001$, $^{***}p < 0.0001$. Underlying images of blots and data can be found in S1 Images and S1 Data, respectively. 10K, 10,000$g$-centrifuged; 100K, 100,000$g$-centrifuged; CNP, 2′,3′-cyclic nucleotide 3′-phosphodiesterase; ND, nutrient deprivation; OL, oligodendroglial; pCN, primary cortical neuron; PLP, proteolipid protein; SEM, standard error of the mean; sEV, small extracellular vesicle; WT, wild-type.

NTA measurement. Remarkably, particle-normalized PLP- and CNP-null sEVs still lacked the ability of wild-type sEVs to rescue starving neurons (Fig 5E). Taken together, these data demonstrate that the reduced rate of sEV secretion in null-mutants likely is sufficient to interfere with sEVs providing neuronal support. Moreover, the qualitative defects in sEV composition also lead to functional impairment, showing that PLP- and CNP-null sEVs are lacking key factors essential for neuronal support. Thus, both qualitative and quantitative defects of mutant sEVs contribute to their malfunction and the failure to promote neuronal viability.

### PLP- and CNP-null sEVs fail to promote axonal transport

To examine the potency of PLP- and CNP-null sEVs to facilitate axonal transport, we performed live imaging of BDNF-mCherry-carrying vesicles in primary hippocampal neurons subjected to nutrient deprivation as described above (Fig 6A). Neurons were treated with sEV-containing supernatants (10K) derived from wild-type, PLP-null, and CNP-null oligodendrocytes and, in addition, from primary cultured astrocytes (all treatment conditions and sEV genotypes were compared within 1 experiment). Astrocytes are known to contribute to neuronal functions in manifold ways, and hence we included this treatment condition as a further control for cell-type-specific sEV action. Intriguingly, treatment with sEVs derived from PLP- and CNP-null oligodendrocytes, as well as sEVs derived from astrocytes, did not rescue fast axonal transport in nutrient-deprived neurons (Fig 6B–6E, please note that control conditions and wild-type sEV data from Fig 2 are reproduced as a reference for the effect of mutant sEVs). Compared to treatment with wild-type sEVs, mutant sEVs reduced both anterograde and retrograde movement of transport vesicles and increased the pool of static vesicles back to the level of untreated nutrient-deprived neurons (Fig 6C). The vesicle velocity was largely unaffected (Fig 6D, reduction of anterograde vesicle velocity by mutant sEVs should be taken with caution due to high variance of measurements). Furthermore, the pausing time of moving vesicles was not reduced by PLP- and CNP-null sEVs (Fig 6E). Thus, the axonal-transport-promoting activity is specific to sEVs derived from oligodendrocytes as opposed to astrocytes. sEVs derived from PLP- and CNP-deficient oligodendrocytes lack this activity and fail to restore axonal transport during nutrient deprivation.

## Discussion

Our study provides new evidence that oligodendrocyte-derived sEVs transported to neurons are critical factors of long-term axonal maintenance. We show that oligodendroglial sEVs confer to neurons the ability to maintain axonal transport under demanding conditions, which is essential for the maintenance of axonal integrity. Consistently, CNP- and PLP-deficient mice with secondary axonal degeneration exhibit impaired sEV release from oligodendrocytes. Mutant-derived sEVs lack the ability to foster the neuronal metabolism and to promote axonal transport. sEV loss of function is accompanied by a common proteomic signature and the reduced abundance of several shared proteins, implicating a set of functionally relevant sEV cargos. Thus, our study establishes sEV delivery from oligodendroglia to neurons as a mechanistic route to sustain neuronal integrity, and provides a so far missing link between glial dysfunction, axonal transport, and axonal degeneration.

Recently, EVs were introduced as signaling entities playing versatile roles in normal brain physiology, during neurodegenerative disease as well as in the context of neuroregeneration [1,2,33]. EVs are highly heterogeneous regarding their origin and cargo, and once present in the extracellular space, it is almost impossible to technically discriminate between plasma-membrane-derived EVs (microvesicles) and endosome-derived EVs (exosomes) [27]. A substantial body of evidence based on electron microscopy and biochemical characterization suggests that oligodendroglial sEVs largely reflect exosomes derived from MVBs [7,8]. In situ, these MVBs frequently appear in the inner cytoplasmic loop next to axons, from where they can be released into the periaxonal space and taken up by axons. We thus assume that exosomes reflect the main functional entity in the sEV preparations with axonal-transport-promoting activity. Consistently, PLP- and CNP-deficient mice exhibit abnormally distributed MVBs in myelinated nerves, suggesting that indeed a defect in exosome release may underlie the axonal degeneration observed in these mice. Based on the present results, we cannot completely rule out that a minor fraction of other sEVs, non-EV particles, or soluble factors

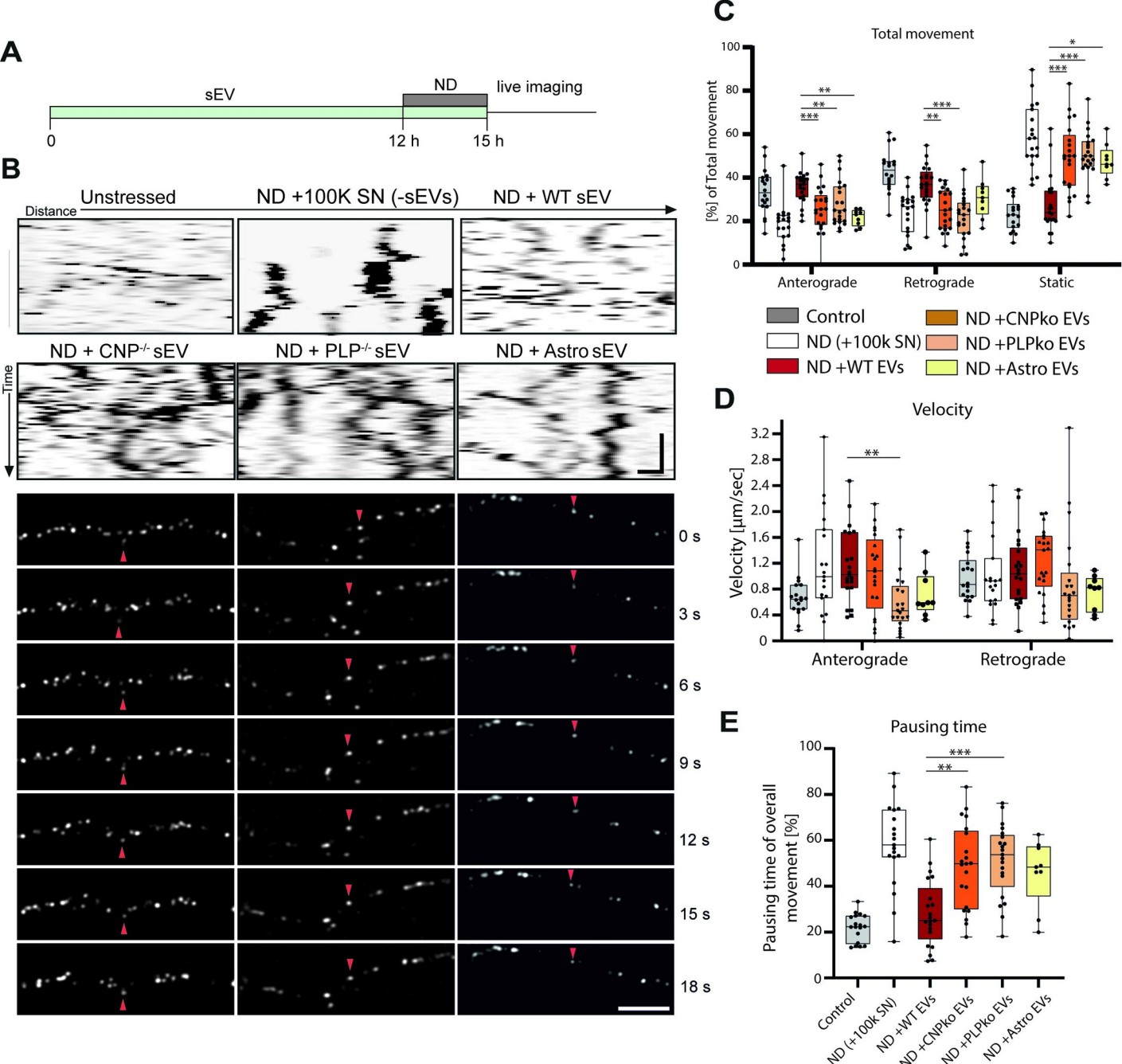

**Fig 6. PLP- and CNP-null exosomes do not promote axonal transport under starvation stress.** (A) Illustration of experimental schedule. Hippocampal neurons were treated with sEVs and subjected to starvation stress by depletion of B27 supplement (nutrient deprivation) during the last 3 h before live imaging. All treatment conditions were identical regarding the presence of oligodendrocyte-derived soluble factors and differed in presence of sEVs. (B) Representative kymographs and corresponding time-lapse frames illustrating movement of BDNF-mCherry-positive vesicles generated from hippocampal neurons exposed to nutrient deprivation and treated with sEVs derived from wild-type (WT), CNP knockout (CNPko), and PLP knockout (PLPko) oligodendrocytes or primary astrocytes (Astro). Red arrowhead follows 1 distinct particle moving over time. Kymograph: horizontal scale bar = 5 μm, vertical scale bar = 10 s; fluorescent picture frames: scale bar = 5 μm. (C–E) Quantitative analysis of kymographs regarding total movement (C), velocity (D), and pausing time (E) of BDNF-mCherry-positive vesicles. Data from control, ND + 100K SN and ND + WT EVs are reproduced from Fig 2 as reference. Data are presented as median, boxes (25th percentile to 75th percentile), and whiskers (minimum to maximum showing all data points), $n = 18$ recorded neurons for control, $n = 19$ for ND + 100K SN (−sEVs); $n = 21$ for ND + WT sEVs, ND + PLPko sEVs, and ND + CNPko sEVs; and $n = 9$ for ND + astrocyte sEVs, with up to 50 vesicle trajectories analyzed per neuron, derived from 3 independent experiments. $^{*}p < 0.05$, $^{**}p < 0.01$, $^{***}p < 0.001$, Shapiro–Wilk normality test following Kruskal–Wallis test with Dunn's multiple comparison test. Significance only indicated for wild-type versus mutant sEVs for reasons of clarity (see Fig 2 for wild-type significance). Underlying data can be found in S1 Data. 100K, 100,000*g*-centrifuged; Astro, astrocyte; CNP, 2′,3′-cyclic nucleotide 3′-phosphodiesterase; ND, nutrient deprivation; OL, oligodendroglial; pCN, primary cortical neuron; PLP, proteolipid protein; SN, supernatant; sEV, small extracellular vesicle; WT, wild-type.

contribute to the reported functional activity of oligodendroglial sEVs. Notably, sEVs isolated from supernatants of primary oligodendrocytes by different methodology including density gradient purification and immuno-bead isolation imply that sEV fractions are homogeneous and poor in non-vesicular particles such as lipoproteins or exomeres previously reported to co-isolate with sEVs [34–36]. Furthermore, the 100K sEV-deprived supernatants employed in functional assays control for other soluble neurotrophic factors that are expected to be secreted by oligodendrocytes [31]. However, it cannot be entirely excluded that soluble factors associating with sEVs in the 100K pellet contribute to the observed effects. Further experiments such as immuno-depletion of specific sEV populations are required to identify the relevant sEV subpopulation and to finally specify the active component within the 10K supernatant.

The finding that oligodendroglial sEVs influence axonal transport provides mechanistic insight into how sEVs, or exosomes, can affect subcellular processes and contribute to overall neural homeostasis. How EVs transmit biological information and execute their functions in target cells is largely unknown [37]. Likewise, we can only speculate about the mode of action by which oligodendroglial sEVs promote axonal transport. According to previous work, neurons endocytose oligodendroglial sEVs even at axonal sites, and the cargo is bioactive upon internalization [8], suggesting that delivered biomolecules may be involved. Moreover, sEVs appear to modulate neuronal signaling pathways by stimulating phosphorylation of Akt, JNK, and GSK-β [9]. Intriguingly, these kinases are known to regulate axonal transport largely by controlling vesicle-associated motor proteins or their adaptors [18,38]. It therefore seems likely that oligodendroglial sEVs directly act on axonal transport via kinase-mediated motor-protein control. Under optimal growth conditions and full energy supplies, we observed shorter transport vesicle pauses shortly after sEV application (30 min) independent of vesicle directionality or velocity, indeed suggesting a kinase-mediated effect on motor activity. In stress experiments, neurons were pretreated with sEVs for several hours before the challenge to mimic persisting sEV availability, as presumed to occur in vivo, where neurons are thought to continuously receive exosomes from neighboring myelinating oligodendrocytes in an activity-dependent fashion. Though the sEV cargo (small molecules, metabolites, or proteins) may well affect axonal transport directly, it is also possible that sEVs exhibit a preconditioning activity (by activation of signaling pathways or by providing protective molecules), allowing the neurons to maintain a homeostatic state even under demanding conditions. However, how sEVs regulate axonal transport on the molecular level and whether sEVs promote axonal transport via kinase regulation or the delivery of specific cargoes (or a combination of both) remains to be established in the future.

Intriguingly, sEV release by oligodendrocytes is impaired in PLP- and CNP-deficient mice, which undergo secondary axonal degeneration, and mutant-derived sEVs lack functional activity. These findings provide compelling genetic evidence that glia-to-neuron exosome transfer is a mode of glial support required for long-term axonal integrity. Glial support is related to supplying neurons with energy-rich metabolites important to satisfy the enormous energy demands associated with neural activity [39,40]. Mature myelinating oligodendrocytes can exist as purely glycolytic cells, and deletion of oligodendroglial monocarboxylate transporter MCT1, delivering lactate and pyruvate, causes axonal damage [14,15]. However, how PLP and CNP deficiency interfere with the delivery of energy substrates has remained less clear so far. Axonal pathology in PLP- and CNP-deficient mice is characterized by the development of axonal swellings filled with organelles, indicating a defect in axonal transport [11,12]. In fact, axonal transport has been shown to be affected in PLP-null mice, while the underlying mechanism remained unknown [13]. Essentially, impaired exosome release in PLP- and CNP-null oligodendrocytes, and the failure of mutant-derived sEVs to promote axonal transport, provides a mechanistic link between glial dysfunction and axonal degeneration. The delivery

of oligodendroglial exosomes to axons, likely concomitant with the supplying of energy substrates, appears to be required to sustain axonal health in the long term. In agreement with this, a recent study demonstrated that secretion of ferritin heavy chain in association with sEVs from oligodendrocytes is required to prevent oxidative damage and neuronal loss in mice [41].

How do PLP and CNP deficiency affect exosome release and function? Based on our observation that MVB number and localization are affected, we assume that the malfunction originating on the glial side results in defective exosome biogenesis and MVB transport. It has been shown that in the absence of CNP the cytoplasmic (myelinic) channels spanning through the myelin sheath are broken down, interfering with cargo transport from the oligodendrocyte cell body to the cytoplasmic loop adjacent to the axon [42]. Consistently, MVBs in CNP-null mice are redistributed to the cell body and less frequently placed at their release site in the periaxonal cytoplasmic loop. Accumulation of MVBs in PLP-null mice may point to a defect in MVB biogenesis or maturation, which could be related to the tetraspan properties of PLP and its ability to bind and recruit cholesterol [43]. In both mutant lines, we observed a numerical decrease in exosome release by roughly 50% and, in addition, qualitative abnormalities in exosome composition, which each by themselves appear sufficient to account for the loss of the neuronal support function and hence augment each other. Quantitative reduction of exosome release is consistent with MVB mislocalization and a lower release rate, while qualitative abnormalities could be associated with either the release from distinct MVB populations or abnormal sorting of exosomal components at the level of the endosome. It is possible that different types of MVBs generate functionally distinct exosome subpopulations that may correlate with their expression profile of the tetraspanins CD63, CD81, and CD9 and other markers [44–46]. However, there is presently no evidence that PLP or CNP deficiency is associated with a shift in MVB subpopulations that would generate exosome pools distinct from wild-type pools. Notably, Alix and Tsg101, which participate in ESCRT-mediated sorting at endosomal membranes, appeared reduced in mutant-derived sEVs, possibly reflecting a general defect in the exosomal sorting machinery.

Remarkably, 5 among the most strongly downregulated exosomal proteins identified by proteomics were identical between PLP and CNP mutant exosomes, thus representing exciting candidates of potential functional relevance. Of note, the chaperone Hsp70-2 and the co-chaperone FKBP1 assist protein folding and thus may be important for axonal proteostasis and neuroprotection [47,48]. This would be consistent with the idea that exosomes deliver macromolecules, including heat shock proteins, conferring stress resistance to axons, which intrinsically lack the ability to modulate the stress response [49]. IGSF8 interacts with CD81 and CD9 in tetraspanin-enriched microdomains and recently was implicated in targeting of perivascular EVs to osteoprogenitor cells, although in this instance IGSF8 was located on the recipient cells [50]. It is well possible that IGSF8 may play a role in targeting oligodendroglial exosomes to neurons. However, the individual functional contribution of these candidate proteins to the support of axonal transport and axonal maintenance needs to be addressed in more detail in future studies to further reveal the molecular mechanisms behind glial support. Moreover, it should be noted that proteins upregulated in PLP- and CNP-null sEVs may exhibit detrimental effects on receiving neurons and impair axonal transport.

In conclusion, our study establishes oligodendrocyte–neuron exosome transfer as a mechanism of glial support—by facilitating of axonal transport—required for long-term axonal maintenance. By this means, metabolic support provided by the supply of energy substrates is complemented by molecular support through the delivery of larger biomolecules via exosomes, maintaining neuronal homeostasis and sustaining axonal projections. Declining axonal transport precedes the axonal degeneration observed in demyelinating diseases and is also a

common hallmark of neurodegeneration [21,23]. Thus, our study provides a previously unrecognized mechanistic link between glial dysfunction and impairment of axonal transport as an initial trigger of axonal damage and may open new avenues to therapeutically interfere with the axon degeneration observed in a broad range of neurodegenerative diseases.

# Materials and methods

## Antibodies and reagents

Antibodies used were as follows: mouse anti-O4 and mouse anti-O1 [51], rat anti-PLP (clone aa3), mouse anti-CNPase (11-5B, Sigma-Aldrich), mouse anti-GFAP (1B4; BD Biosciences), rabbit anti-Iba1 (Wako Chemicals), rabbit anti-Flot-1 (Sigma-Aldrich), mouse anti-AIP1/Alix (49; BD Biosciences), mouse anti-Tsg101 (4A10; GeneTex), mouse anti-Hsp70 (clone 7, BD Biosciences), rabbit anti-IGSF8 (Abcam), rabbit anti-Anxa5 (Abcam), rabbit anti-Sirt2, (Abcam), rabbit anti-hrGFP (StrataGene), rat anti-CD9 (KMC8, BD Biosciences), mouse anti-CD81 (Santa Cruz Biotechnology), rabbit anti-Calnexin (Enzo Life Sciences), rabbit anti-MVP (Abcam), rabbit anti-Histone H3 (Abcam), rat anti-Argonaute 2 (Sigma-Aldrich), mouse anti-MAG (clone 513) carbocyanine and HRP secondary antibodies (Dianova), and Alexa secondary antibodies (Invitrogen).

## Ethics statement

Experiments were in compliance with the German law on animal experimentation, according to the guidelines of the European Union for the use of animals in research (European Union Directive 2010/63/EU). The number of animals sacrificed for the donation of tissue was reported to the German Federal State of Rheinland-Pfalz.

## Animals and cell culture

Mouse strains used were (1) wild type C57Bl/6-N, (2) CNP$^{Cre/Cre}$ (CNCE) [12], and (3) PLP$^{-/-}$ (KPLP) [11,52].

 Primary oligodendrocytes were prepared from embryonic day (E) 14.5 mice as described before [53] or from postnatal day 7–8 mouse brains using the MACS cell separation technology using the Neural Tissue Dissociation Kit and anti-O4 MicroBeads (Miltenyi Biotec) according to the manufacturer's protocol. Briefly, primary oligodendrocytes were isolated from whole brain suspension by magnetic-activated cell sorting via anti-O4 antibodies coupled to magnetic beads and plated at a density of $3 \times 10^6$ cells per 6-cm culture dish. Cells were cultured up to DIV 5 in oligodendrocyte culture medium: NeuroMACS, 20 ml/l NeuroBrew-21 (Miltenyi Biotec), 2 mM L-glutamine, and 10 ml/l 100× Pen-Strep (63.2 μg/ml Penicillin G potassium salt, 135 μg/ml streptomycin sulfate, Gibco). While the MACS cell separation protocol provides cells more rapidly within a few days, both methods result in oligodendrocytes with identical characteristics. To generate oligodendrocyte-conditioned supernatants for application in axonal transport or cell viability assays, O4-sorted oligodendrocytes were kept for 48 h in Neurobasal feeding medium before harvesting and further processing to obtain sEV-containing or sEV-depleted supernatants (see below).

 Primary cortical neurons were prepared from E14.5 brain hemispheres as described before [54]. For primary hippocampal cultures, hippocampi were dissected from brain hemispheres of E18 embryos, treated with 1% trypsin for 4 min at 37˚C, washed in Hank's balanced salt solution containing MgSO$_4$ (HBSS+), and dissociated in pre-warmed DNase by repeated passage through a constricted Pasteur pipette until homogenous. Cell suspension was washed twice by the addition of HBSS+ and subsequent centrifugation at 130$g$ at 4˚C for 10 min.

Cortical and hippocampal neurons were cultured in Neurobasal feeding medium: Neurobasal (Gibco), 20 ml/l B27 (Invitrogen), 0.5 mM L-glutamine, and 10 ml/l 100× Pen-Strep (63.2 μg/ml Penicillin G potassium salt, 135 μg/ml streptomycin sulfate, Gibco). During the first 24 h of culture 12.5 μM glutamate (Sigma-Aldrich) was added to the medium.

Primary astrocytes were derived from E14.5 glial cultures as a byproduct of the primary oligodendrocyte preparation. After shaking off oligodendrocytes from astrocyte monolayers, astrocytes were detached by trypsination (0.01% trypsin, 0.02% EDTA), plated in 6-cm dishes coated with Poly-L-Lysine (Sigma-Aldrich), and grown to confluency.

HEK293T cells were cultured in DMEM + 10% FCS and 1 mM sodium pyruvate.

## Transfection

Hippocampal neurons were transfected with a plasmid encoding BDNF-mCherry (kindly provided by F. Saudou, Grenoble Institut des Neurosciences, Grenoble, France) by nucleofection using the AMAXA Mouse Neuron Nucleofector Kit (Lonza). Each transfection was performed with $1.5 \times 10^6$ hippocampal neurons and plated on 3 video microscopy dishes (ibidi).

## AAV generation, cell infection, and Boyden chamber co-culture

Recombinant AAVs bearing the Cre-recombinase or the hrGFP reporter gene were generated as described previously [8,55]. Primary oligodendrocytes (prepared from E14.5 mice) and cortical neurons were infected by adding 1 μl ($5 \times 10^{7-8}$ vg) to the cell culture medium 6 days before the co-culture experiment. Medium was exchanged 24 h after the transduction to remove excess virus particles. Then, oligodendrocytes (DIV 7, $1.8 \times 10^6$ cells per inset) were co-cultured with cortical neurons ($0.7 \times 10^6$ per well) in Boyden chambers (6-well companion plates [1-μm pores] and 6-well cell culture inserts, BD Falcon). After 24 h, lysates were prepared from the cortical neurons growing in the bottom well.

## sEV isolation by differential ultracentrifugation

Culture supernatants were collected over a period of 48 h under serum-free conditions, and sEVs were isolated by differential centrifugation as described before [8]. Briefly, culture supernatant was cleared of dead cells and debris by successive centrifugation for 10 min at 130$g$ and for 30 min at 10,000$g$ at 4°C in a fixed angle rotor (220.78, Hermle). sEVs remaining in the supernatant were pelleted by ultracentrifugation in polyallomer tubes (Beckman Coulter) for 2 h at 100,000$g$ and 4°C (SW40 rotor, 27,000 rpm, RCF [avg] 92,000$g$, RCF [max] 130,000$g$, k-factor 137 at max speed, Beckman Coulter). sEV pellets were resuspended in PBS for NTA or SDS-PAGE sample buffer for Western blot analysis. Extensive characterization of oligodendroglial sEVs has been performed previously [7–9], and the quality of isolated sEVs was regularly checked by NTA and Western blotting.

## Density gradient centrifugation

For LC-MS analysis, sEVs were subjected to sucrose density gradient centrifugation to further purify exosomes. Briefly, culture supernatants (serum-free) were collected from a total of approximately $2.3 \times 10^8$ cells over 3 subsequent periods of 24 h and subjected to differential centrifugation as described above. In the 100,000$g$ run, sEVs were centrifuged onto a 200-μl sucrose cushion (1.8 M in TBS) for 2 h at 27,000 rpm and 4°C in the SW40 rotor (specifications above). Subsequently, the cushion was diluted and loaded on top of a continuous gradient (0.3–1.8 M sucrose in TBS) followed by centrifugation for 16 h at 27,000 rpm and 4°C in the SW40 rotor (k-factors between 368 and 325 for particles of a density between 1.3 and 1.7 g/

ml, respectively). Density of the fractions was determined using a refractometer (Atago Master-T/2T). Exosome-enriched fractions 5–8 were collected (S4 Fig) and diluted in TBS and centrifuged at 47,000 rpm in a TLA-55 rotor (RCF [avg] 99,000$g$, RCF [max] 136,000$g$, k-factor 66 at max speed, Beckman Coulter) for 1 h. Final exosome pellets were resuspended and directly subjected to trypsin digestion and LC-MS.

High-resolution iodixanol density gradient fractionation was performed as described earlier [34]. In brief, iodixanol (OptiPrep) density medium (Sigma-Aldrich) was prepared in cold PBS immediately before use to generate discontinuous step gradients (12%–36%). Crude 100K pellets of oligodendroglial cell culture supernatant were dissolved in PBS and mixed with iodixanol to a final concentration of 36%. Four iodixanol concentrations (30%, 24%, 18%, 12%) were added stepwise on top of the first layer, yielding the complete gradient. A blank gradient without sample was prepared to determine the density of single fractions. The gradients were subjected to ultracentrifugation at 47,000 rpm at 4°C for 16 h (TLS-55 rotor, RCF [avg] 147,000$g$, RCF [max] 189,000$g$, k-factor 50 at max speed, Beckman Coulter). Twelve individual fractions of 165 µl were carefully collected from the top of the gradient, and density was measured from the empty duplicated gradient using a refractometer (Atago Master-T/2T). Each fraction was diluted with 1 ml of PBS and centrifuged for 2 h at 47,000 rpm at 4°C (TLA-55 rotor, specifications above,) and pellets were resuspended in SDS-PAGE sample buffer for further Western blot analysis.

## Size exclusion chromatography

EV isolation by size exclusion chromatography (SEC) was implemented using self-made columns as described [56]. Briefly, Sepharose CL2B (Sigma-Aldrich) columns were packed in a 10-ml syringe, with a 0.45-µm PES filter membrane at the bottom of the syringe. After washing of the column with PBS, 2 ml of 10,000$g$-centrifuged oligodendroglial cell culture supernatant concentrated by Amicon Ultra-15 (100-kDa cutoff) was loaded on the column, and 1-ml fractions were collected. Single fractions were centrifuged for 2 h at 47,000 rpm at 4°C (TLA-55 rotor, specifications above), and pellets were resuspended in SDS-PAGE sample buffer for further Western blot analysis.

## Immunobead isolation

EV preparation by immuno-bead isolation was performed using the Pan (mixture of CD9, CD63, and CD81) Exosome Isolation Kit according to the manufacturer's protocol (Miltenyi Biotec). Then 2 ml of Amicon Ultra-15 (100-kDa cutoff) concentrated 10,000$g$-centrifuged oligodendroglial cell culture supernatant was used as starting material.

## Nanoparticle tracking (NTA)

sEVs (100K pellets) derived from $1.2 \times 10^7$ oligodendrocytes were resuspended in PBS and analyzed using the NanoSight LM10 system equipped with the green laser (532 nm) and the syringe pump and NanoSight 2.3 software (Malvern) at 23°C (temperature control). The following settings were used: camera control in standard mode (camera level 16) and particle detection in standard mode (screen gain 16, detection threshold 6, and minimum expected particle size auto). Script control was used (Repeatstart, Syringeload 500, Delay 5, Syringestop, Delay 15, Capture 30, Repeat 4). Five 30-s videos were recorded, particles were tracked (batch process), and average values were formed. Particle concentration was related to the volume of culture supernatant (particles/milliliter) or the number of secreting cells (particles/cell).

### sEV treatment conditions (nutrient deprivation and oxidative stress)

For axonal transport experiments and MTT assays conducted under nutrient deprivation, sEV-containing 10K supernatants or sEV-depleted 100K supernatants lacking the B27 supplement were applied to neurons. For experiments under normal or oxidative stress conditions, sEVs comprising exosomes were resuspended in PBS before being applied to neurons. sEV normalization was performed according to the number of sEV-producing cells or the number of particles determined by NTA (MTT assays). In the case of control sEVs derived from HEK293T cells or primary astrocytes (which are much larger cells compared to oligodendrocytes), sEV input was according to the same volume of supernatant harvested from confluent cells covering the same area.

### Axonal transport assay

Imaging of axonal transport was performed on BDNF-mCherry transfected primary hippocampal neurons. sEV treatment, oxidative stress, and nutrient deprivation conditions were performed according to [8]. sEV input was normalized according to the numbers of secreting cells: sEVs derived from $3 \times 10^6$ sEV-secreting oligodendrocytes were added to $0.5 \times 10^6$ neurons. Nutrient deprivation of neurons was performed by depletion of B27 supplement from the culture medium. Before nutrient deprivation, neurons were pre-incubated with oligodendrocyte-conditioned culture supernatant containing sEVs and B27 supplement (10K supernatant) for 12 h, followed by a medium exchange to B27-depleted, sEV-containing 10K supernatant for 3 h. Control cells were treated accordingly with oligodendrocyte-conditioned 100K supernatant depleted of sEVs.

Unstressed hippocampal neurons were treated with 100K pelleted sEVs 30 min before live cell imaging. For imaging axonal transport under oxidative stress conditions, neurons were treated with 100K pelleted sEVs for 12 h followed by addition of 15 μM $H_2O_2$ (Roth) for 1 h and live cell imaging. Control cells received an equal volume of PBS before stress exposure.

### Video microscopy and quantification of axonal transport

Axons of hippocampal neurons were imaged at DIV 2 or DIV 3 after isolation and transfection using a Zeiss Axiovert 200M microscope equipped with a 63× oil-immersion lens (untreated and oxidative stress experiments) or a Leica TCS SP5 microscope equipped with a 100× oil-immersion lens (nutrient deprivation experiments). During imaging, cells were kept at 37°C and 5% $CO_2$. Imaged neurons were in an early maturation stage representing stage 3 of axon determination. Axons were identified by defined morphological criteria according to Dotti et al. [29]: Axons are 1.5 times longer than the longest neurite. The images were generated from 3 to 5 μm far away from the cell body and 3 to 5 μm before the axonal growth cone. Time-lapse recordings were acquired by scanning single plane images every 0.5 s (nutrient deprivation assays) or 3 s (full medium and oxidative stress experiments), for a total of 200 frames.

To quantify parameters of axonal transport (anterograde and retrograde movement, velocity, pausing time), kymographs from processed movies were generated using the ImageJ software with the Multi Kymograph plugin. Up to 50 trajectories per kymograph were analyzed (1 trajectory equals 1 particle). Trajectories were classified in anterograde or retrograde movement if they showed positive or negative velocity, respectively. Average velocities of single vesicles were calculated as (final distance − initial distance)/(final time − initial time) for each vesicle trace. Vesicles with a mean velocity below 5 μm/min (0.083 μm/s) were considered static. A pause was defined as a period ≥ 10 s during which the vesicle moved less than 5 μm. The cutoff of 5 μm was chosen based on the average diameter of vesicles, which was

approximately 1 μm. Pausing time is the percentage of time that a vesicle spends pausing over its total time movement. We quantified 0 to 7 pausing vesicles per axon (modified from [57]).

### Cell viability/metabolic activity assay

Cortical neurons seeded at a density of $7.4 \times 10^4/cm^2$ were treated in the absence of B27 supplement for 15–17 h with oligodendrocyte-conditioned 10K or 100K supernatants, which contain sEVs or are depleted of sEVs, respectively. Despite the presence or absence of sEVs, these supernatants are identical regarding other secreted oligodendroglial factors that potentially could influence neurons. For normalization, sEVs derived from an equal number of producing cells ($3 \times 10^6$ cells per treatment condition) or an equal number of particles as determined by NTA were applied to neurons. For dose–response analysis, 10K supernatants ($2.48 \times 10^8 \pm 0.12 \times 10^8$ particles/ml = 100%) were diluted in medium to a relative concentration of 80%, 60%, 40%, and 20% or depleted of sEVs (100K = 0%). To normalize particle concentration between wild-type, CNP-null, and PLP-null sEVs, 10K supernatants were concentrated by ultrafiltration (Amicon, 100-kD filter), and particle concentration was adjusted according to NTA measurement to $2.6 \times 10^8$ particles/ml for treatment of nutrient-deprived neurons (743 particles/neuron). Metabolic activity of neurons was determined by the MTT assay. Briefly, 0.75 mg/ml MTT (Sigma-Aldrich) was added to culture medium for 2 h. Formazan crystals formed were solubilized in a buffer containing 40% (v/v) dimethyl-formamide (Sigma-Aldrich), 10% (w/v) SDS, and 2% (v/v) acetic acid overnight. The absorbance was measured at 562 nm using a plate reader (Tecan Infinite M200 PRO).

### Cell lysates, Western blotting, and immunocytochemistry

Cells were scraped in 10 mM Tris (pH 7.4), 150 mM NaCl, 1 mM EDTA, 1% Triton X-100 and protease inhibitor cocktail (Roche complete) on ice. Nuclei were pelleted by centrifugation for 10 min at 300$g$. Cell lysates and sEV/exosome samples (derived from $1.6 \times 10^7$ cells) were subjected to 12% SDS-PAGE and Western blotting (Bio-Rad) or to 4%–12% Bis-Tris gel and Western blotting (NuPAGE, Life Technologies). Proteins were blotted onto a PVDF membrane, which was subsequently blocked with 4% milk powder and 0.1% Tween in PBS. Membranes were sequentially incubated with primary and HRP-coupled secondary antibodies, and proteins were detected using chemiluminescence reagents (Luminata Crescendo, Millipore) and X-ray films. Films were scanned and analyzed using ImageJ software (National Institutes of Health). Immunocytochemical staining of cells was performed as described [54]. Fluorescence images were acquired using a fluorescence microscope (DM6000m, Leica) and processed with Image J.

### Electron microscopy

To capture exosomes released by primary oligodendrocytes, cells were fixed in 4% formaldehyde and 0.2% glutaraldehyde in 0.1 M phosphate buffer in the dish and processed as described [7]. Briefly, cells were scraped in 0.1 M phosphate buffer containing 1% gelatin, pelleted, and resuspended in 10% gelatin in 0.1 M phosphate buffer at 37˚ C. Gelatin pellets were cut in small blocks, infiltrated in 2.3 M sucrose in 0.1 M phosphate buffer, mounted onto aluminum pins for ultramicrotomy, and frozen in liquid nitrogen. Ultrathin cryosections were picked up in a 1:1 mixture of 2% methylcellulose and 2.3 M sucrose. For immuno-labeling, sections were incubated with 513 antibodies specific for MAG, which was detected with protein Agold (10 nm). Sections were analyzed with a Leo EM912 Omega electron microscope (Zeiss, Oberkochen, Germany), and digital micrographs were obtained with an on-axis 204862048-pixel CCD camera (TRS).

For ultrastructural analysis of MVBs, adult mice (age 10 weeks) were fixed by perfusion with 4% formaldehyde (Serva) and 0.2% glutaraldehyde (Science Services) in 0.1 M phosphate buffer containing 0.5% NaCl. Gelatin-embedded pieces of optic nerves were infiltrated in 2.3 M sucrose in 0.1 M phosphate buffer overnight. Pieces of optic nerve samples were mounted onto aluminum pins for ultramicrotomy and frozen in liquid nitrogen. Ultrathin cryosections were prepared using a cryo-ultramicrotome (UC6 equipped with a FC6 cryobox, Leica). Sections were analyzed as described above. To quantitate MVBs, cryosections of optic nerves from 3 different animals per genotype were analyzed. Ten images were taken from every animal at 8,000× magnification, each covering an area of 11.5 μm × 11.5 μm, summing up to a total area of 1,322.5 $\mu m^3$ per animal. On every image the number and localization of MVBs were evaluated, and the total number of myelinated axons was counted. The number of MVBs is calculated relative to the number of axons in the imaged field. For analyses, the mean of the 10 images per animal (3 animals per genotype) was calculated and compared between different genotypes.

## LC-MS

For MS analysis, sEVs were collected from a total of $2.3 \times 10^8$ primary oligodendrocytes (>3 independent cell preparations, each derived from a pool of >10 individual embryonic mouse brains) and enriched for exosomes by successive differential centrifugation and sucrose density gradient centrifugation as described above. Density-gradient-purified exosomes were pelleted in 1.5-ml LoBind Eppendorf tubes and solubilized in 50 mM ammonium bicarbonate, 0.2% RapiGest (Waters, Eschborn, Germany). Proteins were reduced by adding 5 mM DTT (45 min, 56˚C) and free cysteines alkylated with iodoacetamide (Sigma-Aldrich, Taufkirchen, Germany; 15 mM, 25˚C, 1 h in dark). Sequencing-grade trypsin (Promega, Mannheim, Germany) was added, and the samples were incubated overnight at 37˚C. After digestion, RapiGest was hydrolyzed by adding 10 mM HCl (37˚C, 10 min), and the resulting precipitate was removed by centrifugation (13,000*g*, 15 min, 4˚C). The supernatant was transferred into an autosampler vial for peptide analysis by LC-MS.

Nanoscale LC separation of tryptic peptides was performed with a nanoACQUITY system (Waters) equipped with an ethylene bridged hybrid (BEH) C18 analytical reversed-phase column (1.7 μm; 75 μm by 150 mm) (Waters) in direct injection mode. Two microliters of tryptic digest was injected per technical replicate. Mobile phase A was water containing 0.1% (vol/vol) formic acid, while mobile phase B was acetonitrile (ACN) containing 0.1% (vol/vol) formic acid. The peptides were separated with a gradient of 3% to 40% mobile phase B over 60 min at a flow rate of 300 nl/min, followed by a 10-min column rinse with 90% mobile phase B. The columns were reequilibrated under the initial conditions for 15 min. The analytical column temperature was maintained at 55˚C. The lock mass compound, [Glu1]-Fibrinopeptide B (500 fmol/μl), was delivered by the auxiliary pump of the LC system at 300 nl/min to the reference sprayer of the NanoLockSpray source of the mass spectrometer.

Mass spectrometric analysis of tryptic peptides was performed using a Q-Tof Premier mass spectrometer (Waters). For all measurements, the mass spectrometer was operated in V-mode with a typical resolution of at least 10,000 FWHM (full width half maximum). All analyses were performed in positive-mode electrospray ionization (ESI). The time of flight analyzer of the mass spectrometer was externally calibrated with a NaI mixture from m/z 50 to 1,990. The data were acquisition lock mass corrected using the doubly charged monoisotopic ion of [Glu1]-Fibrinopeptide B. The reference sprayer was sampled with a frequency of 30 s. Accurate LC-MS data were collected in data-independent modes of analysis. The spectral acquisition time in each mode was 0.7 s, with a 0.05-s interscan delay. In low-energy MS mode, data were

collected at a constant collision energy of 4 eV. In elevated-energy MS mode, the collision energy was ramped from 25 to 55 eV during each 0.6-s integration. One cycle of low- and elevated-energy data was acquired every 1.5 s. The radio frequency amplitude applied to the quadrupole mass analyzer was adjusted so that ions from m/z 350 to 2,000 were efficiently transmitted, ensuring that any ions observed in the LC-MS data less than m/z 350 were known to arise from dissociations in the collision cell. All samples were analyzed in quintuplicate.

## Data processing

Continuum LC-MS data were processed and searched using ProteinLynx Global SERVER version 3.0.2 (Waters). Protein identifications were obtained by searching a custom compiled database containing sequences of the UniProt/Swiss-Prot Mouse reference proteome database (17,005 entries). Sequence information of enolase 1 (*Saccharomyces cerevisiae*) and bovine trypsin was added to the databases to normalize the datasets or to conduct absolute quantification as described previously [58]. Guideline identification criteria were applied for all searches. The LC-MS data were searched with a 10-ppm precursor and 20-ppm product ion tolerance, with 1 missed cleavage allowed and fixed carbamidomethylcysteine and variable methionine oxidation set as the modifications.

## Statistical analysis

Quantitative data are shown as median of at least 3 independent experiments. Values are illustrated with boxes and whiskers, with a horizontal line depicting the median, boxes showing 25th percentile to 75th percentile, and whiskers showing minimum to maximum with all data points. Statistical analysis was performed when $n > 4$ using GraphPad Prism Software X8. Data were subjected to the Shapiro–Wilk normality test [59]. Values evaluated in axonal transport experiments were not normally distributed and were analyzed using a non-parametric, unpaired Kruskal–Wallis test with Dunn's multiple comparison test (nutrient deprivation [Figs 2 and 6], oxidative stress [S2 Fig]) or using a 2-tailed, non-parametric, unpaired Mann–Whitney test (resting conditions [S1 Fig]).

Data derived from MTT assays comparing neuronal metabolic activity in response to wild-type and mutant sEVs were analyzed by 2-tailed, unpaired Student *t* test.

## EV-TRACK

We have submitted all relevant data of our experiments to the EV-TRACK knowledgebase (https://evtrack.org/) (EV-TRACK ID: EV200182) [60].

## Supporting information

**S1 Data. Quantitative data obtained by the different assays reported in the paper.** (XLSX)

**S2 Data. Quantitative proteomics data.** (XLSX)

**S1 Fig. Influence of oligodendroglial sEVs on axonal transport under resting conditions.** (A) Illustration of experimental schedule. sEVs were applied to hippocampal neurons and imaged after 30 min. (B) Representative kymographs and corresponding time-lapse frames illustrating movement of BDNF-mCherry-positive vesicles along the axon of primary hippocampal neurons. Neurons were sham treated (control) or treated with oligodendroglial (OL) sEVs. Red arrowhead follows 1 distinct particle moving over time. Kymograph: horizontal

scale bar = 5 μm, vertical scale bar = 1 min; frames: scale bar = 5 μm. (C–E) Quantitative analysis of kymographs considering total movement (C), velocity (D), and pausing time (E) of BDNF-mCherry-positive vesicles. Data are presented as median, boxes (25th percentile to 75th percentile), and whiskers (minimum to maxamium showing all data points), $n$ = 12 recorded neurons from 3 independent experiments. $^{**}p < 0.01$, Shapiro–Wilk normality test and 2-tailed, non-parametric, unpaired Mann–Whitney test. Underlying data can be found in S1 Data.
(TIF)

**S2 Fig. Influence of sEVs on axonal transport under oxidative stress.** (A) Illustration of experimental schedule. Hippocampal neurons were pretreated with sEVs before exposure to oxidative stress (OS) and live imaging. (B) Representative kymographs and corresponding time-lapse frames illustrating movement of BDNF-mCherry-positive vesicles generated from hippocampal neurons sham treated or treated with sEVs derived from oligodendrocytes (OL) or HEK293T cells. Red arrowhead follows 1 distinct particle moving over time. Kymograph: horizontal scale bar = 5 μm, vertical scale bar = 1 min; frames: scale bar = 5 μm. (C–E) Quantitative analysis of kymographs regarding total movement (C), velocity (D), and pausing time (E) of BDNF-mCherry-positive vesicles. Data are presented as median, boxes (25th percentile to 75th percentile), and whiskers (minimum to maxamium showing all data points), $n$ = 13 for untreated and OL-sEV-treated neurons and $n$ = 7 for HEK-sEV-treated neurons derived from 3 independent experiments. $^{*}p < 0.05$, $^{**}p < 0.01$, $^{***}p < 0.001$, Shapiro–Wilk normality test following non-parametric, unpaired Kruskal–Wallis test with Dunn's multiple comparison test. Underlying data can be found in S1 Data.
(TIF)

**S3 Fig. Normal development and viability of PLP- and CNP-null oligodendrocytes.** Wild-type, PLP-null, and CNP-null oligodendrocytes were compared regarding their differentiation and cell viability. (A) Immunofluorescence staining of primary cultured oligodendrocytes after 6 days in vitro (DIV) using antibodies against PLP and CNP as well as differentiation markers MOG, O4, and O1. Morphological appearance and differentiation of mutant oligodendrocytes appear normal. (B) Western blot analysis of wild-type and mutant oligodendrocytes after 1, 5, and 7 DIV using differentiation markers (NG2, PLP/DM20, CNP, MOG, MBP) as well as Rab35, which is a GTPase regulating sEV/exosome release. GAPDH serves as loading control. (C) MTT cell viability assay conducted at 6 DIV, $n$ = 8. (D) Lactate dehydrogenase (LDH) cytotoxicity assay to determine cell death in the cultures. At day 6 in culture, primary oligodendrocytes were incubated with fresh medium for 24 h and subjected to LDH assay (Roche, carried out according to the manufacturer's protocol), $n$ = 8. Underlying images of blots and data can be found in S1 Images and S1 Data, respectively.
(TIF)

**S4 Fig. Density gradient analysis of wild-type, CNP-null, and PLP-null sEVs.** sEVs isolated from culture supernatants derived from equal numbers of wild-type, CNP-null, and PLP-null oligodendrocytes by differential ultracentrifugation followed by density gradient centrifugation (0.3–1.8 M sucrose gradient). Individual fractions were analyzed by Western blotting using established oligodendroglial sEV markers, PLP/DM20 (DM20 is a smaller isoform of PLP), Sirtuin-2, and Flotillin-1. Slight shifts in fractions between the markers indicate heterogeneity in sEV/exosome populations, which has been recognized previously [8]. Wild-type and mutant sEVs appear of similar density. Note that marker intensity appears weaker in mutant sEVs, reflecting a lower sEV yield from the same volume of starting material.

Underlying images of blots can be found in S1 Images.
(TIF)

**S5 Fig. Clustered heatmap analysis of the wild-type, CNP-null, and PLP-null proteome.**
Relative expression values for all quantified proteins were used for an unsupervised hierarchical cluster analysis using the ClustVis Toolbox (PMID: 25969447). The analysis indicates clear separation of EVs from the different genetic backgrounds. Underlying data can be found in S2 Data.
(TIF)

**S6 Fig. STRING analysis of proteins downregulated in CNP-null versus wild-type sEVs.**
Details can be found at https://version-11-0.string-db.org/cgi/network.pl?networkId=aIjAq7kLHD14.
(TIF)

**S7 Fig. STRING analysis of proteins upregulated in CNP-null versus wild-type sEVs.**
Details can be found at https://version-11-0.string-db.org/cgi/network.pl?networkId=94bMLC7bXyTe.
(TIF)

**S8 Fig. STRING analysis of proteins downregulated in PLP-null versus wild-type sEVs.**
Details can be found at https://version-11-0.string-db.org/cgi/network.pl?networkId=qPmNBslbsv6T.
(TIF)

**S9 Fig. STRING analysis of proteins upregulated in PLP-null versus wild-type sEVs.** Details can be found at https://version-11-0.string-db.org/cgi/network.pl?networkId=jewb5zs2ixkl.
(TIF)

**S1 Images. Images of uncropped Western blots.**
(PDF)

## Acknowledgments

We thank Lilja Niedens for technical help, Sandra Ritz for providing expertise on live cell imaging and image analysis, and Dr. Elmo Neuberger for advice regarding statistical analysis. We are grateful to A. Andrieux for hosting WPKE in her laboratory and for valuable discussion.

## Author Contributions

**Conceptualization:** Carsten Frühbeis, Wen Ping Kuo-Elsner, Eva-Maria Krämer-Albers.

**Data curation:** Carsten Frühbeis, Wen Ping Kuo-Elsner, Christina Müller, Kerstin Barth, Stefan Tenzer, Dominik Fröhlich, Eva-Maria Krämer-Albers.

**Formal analysis:** Carsten Frühbeis, Wen Ping Kuo-Elsner, Christina Müller, Kerstin Barth, Leticia Peris, Wiebke Möbius, Dominik Fröhlich, Eva-Maria Krämer-Albers.

**Funding acquisition:** Eva-Maria Krämer-Albers.

**Investigation:** Carsten Frühbeis, Wen Ping Kuo-Elsner, Christina Müller, Kerstin Barth, Stefan Tenzer, Wiebke Möbius, Dominik Fröhlich.

**Methodology:** Carsten Frühbeis, Wen Ping Kuo-Elsner, Kerstin Barth, Leticia Peris, Stefan Tenzer, Wiebke Möbius, Eva-Maria Krämer-Albers.

**Project administration:** Carsten Frühbeis, Eva-Maria Krämer-Albers.

**Resources:** Stefan Tenzer, Hauke B. Werner, Klaus-Armin Nave.

**Software:** Stefan Tenzer.

**Supervision:** Eva-Maria Krämer-Albers.

**Validation:** Carsten Frühbeis, Wen Ping Kuo-Elsner, Christina Müller, Stefan Tenzer.

**Visualization:** Carsten Frühbeis, Wen Ping Kuo-Elsner, Christina Müller, Kerstin Barth, Stefan Tenzer.

**Writing – original draft:** Carsten Frühbeis, Wen Ping Kuo-Elsner, Eva-Maria Krämer-Albers.

**Writing – review & editing:** Eva-Maria Krämer-Albers.

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
