## [Editor Report · Decision Letter 0]

17 Dec 2019

Dear Dr Krämer-Albers, 

Thank you for submitting your manuscript entitled "Oligodendrocyte-derived exosomes promote axonal transport and axonal long-term maintenance" for consideration as a Research Article by PLOS Biology.

Your manuscript has now been evaluated by the PLOS Biology editorial staff as well as by an academic editor with relevant expertise and I am writing to let you know that we would like to send your submission out for external peer review.

Please re-submit your manuscript within two working days, i.e. by Dec 19 2019 11:59PM.

***Please be aware that, due to the voluntary nature of our reviewers and academic editors, manuscripts may be subject to delays due to their limited availability during the holiday season. Please also note that the journal office will be closed entirely 21st- 29th December inclusive, and 1st January 2020. Thank you for your patience.***

Kind regards,

Ines

--

Ines Alvarez-Garcia, PhD

Senior Editor

PLOS Biology

Carlyle House, Carlyle Road

Cambridge, CB4 3DN

+44 1223–442810

---

## [Decision Letter · Decision Letter 1]

24 Jan 2020

Dear Dr Krämer-Albers,

Thank you very much for submitting your manuscript "Oligodendrocyte-derived exosomes promote axonal transport and axonal long-term maintenance" for consideration as a Research Article at PLOS Biology. Thank you also for your patience as we completed our editorial process, and please accept my apologies for the delay in providing you with our decision. Your manuscript has been evaluated by the PLOS Biology editors, an Academic Editor with relevant expertise, and by four independent reviewers.

As you will see, the reviewers find the conclusions of your manuscript novel and interesting, however they also raise several concerns that need to be addressed before we can consider further your manuscript for publication. After discussing the reviews with the Academic Editor, we do feel that Reviewer 2's Point 4 will be very difficult to address without knowledge of OL-sEV proteins that are otherwise completely absent from neurons, thus you can just address this point by adding a discussion in the text. Reviewer 3 brings up valid points regarding whether or not axons are fully specified in these cultures at the time of the assay and potential issues with the imaging methods, which we do feel need to be addressed experimentally. The question regarding the morphology of CNP and PLP null neurons is interesting, but we feel it is beyond the scope of this study and we won't make it essential for publication.

In light of the reviews (attached below), we will not be able to accept the current version of the manuscript, but we would welcome re-submission of a revised version that takes into account the reviewers' comments. We cannot make any decision about publication until we have seen the revised manuscript and your response to the reviewers' comments. Your revised manuscript is also likely to be sent for further evaluation by the reviewers.

We expect to receive your revised manuscript within 2 months. 

**IMPORTANT - SUBMITTING YOUR REVISION**

*NOTE: In your point by point response to to the reviewers, please provide the full context of each review. Do not selectively quote paragraphs or sentences to reply to. The entire set of reviewer comments should be present in full and each specific point should be responded to individually, point by point.

*Re-submission Checklist*

*Published Peer Review*

*PLOS Data Policy*

*Blot and Gel Data Policy*

Sincerely,

Ines

--

Ines Alvarez-Garcia, PhD

Senior Editor

PLOS Biology

Carlyle House, Carlyle Road

Cambridge, CB4 3DN

+44 1223–442810

Reviewers’ comments

Rev. 1: Graca Raposo – note that this reviewer has waived anonymity.

In this manuscript Frühbeis and collaborators investigate how the intricate intercellular communication between glial cells and neurons can ensure long-term integrity of neuronal axons. In particular they investigate in detail how small Extracellular Vesicles with features of exosomes released by oligodendrocytes can support neurons and axonal transport even in stress conditions (oxidative stress, nutrient deprivation).

The team has previously reported how communication within the CNS can largely rely on a loop on intercellular communication based on release and transfer of exosomes in between glial cells and neurons. The present study extends these initial observations and brings light on how EV scan also support axonal integrity.

In my opinion the manuscript is very complete and detailed with a lot of appropriate controls. The results support the conclusions and certainly sets novel concepts in the understanding of how such loop of intercellular communication is also important for axonal maintenance and have implications for molecular basis of neurodegenerative diseases.

In the moving field of Extracellular Vesicles, the authors are very cautious with the nomenclature and it is very clear for non-advised readers in the field.

I have only 2 minor points:

The authors have previous characterized the EVs secreted by oligodendrocytes, although it would be may useful in the manuscript to include a small supplement with the minimal characterization of the vesicles?

The authors show, using different methods a reduction, both quantitative and qualitative, of sEVs derived from CNP and PLP-null oligodendrocytes. They focused then on downregulated candidates though it is not very clear why the levels of TSP CD63/CD81 are not altered if the numbers of sEVs are reduced?

Are we dealing with different MVB subpopulations that are affected in the null PLP and null CNP?

I am unsure whether this point will be clear for all readers.

Rev. 2:

The manuscript "Oligodendrocyte-derived exosomes promote axonal transport and axonal long term Maintenance" by Frühbeis et al described an interesting observation that oligodendroglial exosomes support neurons by promoting fast axonal transport, especially under conditions of oxidative stress and nutrient deprivation, which represents a potentially interesting and new mechanism how oligodendroglia support axonal health. The authors also explored the quantitative and cargo changes of mutant oligodendroglia-derived exosomes and concluded that these changes in exosomes is one of the possible mechanisms underlying glia-dependent secondary axonal degeneration in these mutant mice. However, this study, at its current format, suffers several major weaknesses, including design flaws, incompletion in certain experiments; confusing organization of several figures with unclear legend, and missing of important controls, which significantly undermine the conclusion.

Major comments:

1. The conclusion that oligodendrocytes-derived sEV promotes axonal transport is not convincing in Figure. 1. Most of the results presented in this figure is not different. The significance of pausing time on the axonal transport and health is not clearly described. In addition, although different fractions such as 10K or 100K supernatant was used in the treatment in other Figures. These fractions were not used in this experiment, making it unclear how the baseline effect of these fractions is. In addition, the important control, the nutrient deprivation (ND) alone is lacking, making it difficult to assess the effect of different fractions on these neurons. Moreover, the use of ND in the manuscript is misleading, as it refers to the treatment of 100K supernatant from oligodendrocyte conditioned medium, but not the nutrient deprivation alone. It is important to recognize that there are other factors in oligodendrocyte conditioned medium that promotes neuronal survival, such as Wilkins A et al., 2003, J of Neuroscience, 23 (12): 4967-4974.

2. Although ultracentrifugation has been widely used to purify exosomes, it has always been criticized for having proteins especially lipoproteins bound or co-pelleted on the outside membrane, such as Li P et al (Theranostics, 2017, Vol 7, 3, 789-804). This issue was not sufficiently addressed in this study. The authors compared the effect of 10K and 100K supernatant and concluded that oligodendrocytes-derived sEV is protective. It is highly likely that a supernatant protein is attached to the outside membrane of sEV and co-pelleted during the 100K spin and be neuroprotective. Additional experiments to rule out this possibility is important to support the authors' claim. Also, why the proteomics and treatment uses different approach to purify exosomes? There is no data presented to indicate that the gradient centrifugation purified sEVs are protective in these assays.

3. The study primarily focus on the axon transport to assess axon health. As the secondary axonal degeneration is the main phenotype in CNP- and PLP-deficient mice, it will be more relevant to test whether sEVs from oligodendrocytes are affecting axon growth, conductivity, and neuronal survival, etc. This is important to rule out the possibility that SNP- and PLP-deficient oligodendrocytes are not secreting toxic sEVs or other factors that negatively affects the survival of neurons and subsequently affects the axon transport. Especially neurons under oxidative stress in Figure 2 showed much reduced axon transport property.

4. The cre-mediated uptake assay of sEV is not convincing. The proteomics results have suggested that the cargos are altered in CNP- and PLP-deficient oligodendrocytes-derived sEVs. Thus, it is very likely that Cre-packaging into sEV is also altered in CNP- and PLP-deficient oligodendrocytes-derived sEVs, contributing to a reduced GFP activation in neurons. However, this may not indicate reduced uptake of sEVs into neurons. These results need to be further confirmed in neurons using additional protein cargos identified in proteomics.

Minor comments:

1. Figure legend is misleading in describing the ND condition, which makes relevant figures confusing.

2. Immuno-EM should be performed to specifically label MVBs instead of the morphological description.

3. Information missing as to the number of sections or mice examined for MVB quantification.

Rev. 3: Tomas Falzone – note that this reviewer has waived anonymity

This work entitled "Oligodendrocyte-derived exosomes promote axonal transport and axonal long-term Maintenance" by Frühbeis C, et al. is focused on understanding the possible role of extracellular vesicles released by oligopdendrocites in the regulation of transport dynamics and in long term neuronal survival. This question has significant relevance in the process of neurodegeneration and regeneration. To lay out this proposition Frühbeis uses primary neuronal cultures transfected with a fluorescent cargo (BDNF) and incubate them with extracellular vesicles obtained from oligodendrocytes (O-EVs). This gave mild effects on reducing the pausing time of BDNF. Then authors showed that O-EVs were able to prevent some transport defects induced by oxidative stress or nutrient deprivation. Authors also studied how two mutant mouse lines (PLP- and CNP-deficient mice), which are established mouse models of glia-dependent secondary axonal degeneration, affect the production of O-EVs. The quantity of O-EVs release and the protein conformation within released O-EVS is impaired in these mutant mice. Downrgulation and upregulatin of protein content within O-EVs was determined by mass-spec and candidates were validated by western blot. In addition, mutant O-EVs failed to deliver CRE enzyme activity in a functional test Finally authors perform experiments to show that WT O-EVs are sufficient to restore axonal transport from nutrient deprived conditions, however EVs obtained from mutant cells do not produce this effect. The proposition that normal O-EVs regulate axonal transport properties is novel and very interesting. Also that mutant O-EVs lack key protein components and fail to promote transport under nutrient deprived conditions. However, I see within the manuscript two sets of non-cohesive group of experiments and many experimental settings and statistical conflicts that need to be solved before considering this work for publication.

Major concerns

* The title does not reflect the relevance of exosome variations in mutant mice. The main problem is that the main questions, regarding transport dependence on O-EVs and defects in mutant O-EVs release are not well resolved. This work would be greatly improved with more cohesiveness regarding both controls and experimental design. As well as a clear focus in answering the most relevant questions that authors themselves presented.

* Experimental setting should be standardized in order to compare different results in different figures properly. Treatment time frames differ from Fig 1 (30 min) of O-EVs incubation to Fig 2-3 (15 hours), this should be explained.

* In fig 1, authors showed: "Primary hippocampal neurons expressing BDNF-mCherry were exposed to oligodendrocyte-derived sEVs". In Fig 3 authors showed :"Primary hippocampal neurons were treated with sEV-conditioned oligodendrocyte supernatant (10K) or sEV-deprived supernatant (100K) as control". This change in methodology and the possibility that affect neurons differently should be discussed mainly due to the changes that are observed in one experiment compared to the other.

* Imaging experiments in electroporated primary hippocampal neurons were performed and imaged at: "at DIV 2 or DIV 3 after isolation". This raises a significant concern about whether the authors are truly observing axonal transport or they are just looking at intracellular dynamics in undefined projections. It has been thoroughly demonstrated that axon determination start between day 2 to 4 in culture of hippocampal neurons (Dotti, journal of Neuroscience 1988). Method for imaging are incomplete. How authors defined the axon? Where in the axon between cell body and axonal tip imaging was generated? Was microtubule polarity defined in order to identify the axon projection and maturation?

* Why authors performed time lapse movies every 3 seconds to register a fast moving vesicle? In many of the kymographs showed in figures it is very difficult to follow the vesicle trajectory.

* A major concerns is placed in the identification of 40 um/sec velocities or above. Since authors performed time lapse movies every 3 seconds particles should disappear from view of field in this time frame. Some velocity conversion does not fit with previous velocities described for BDNF that range at around 1 um/second (Olenick, Journal of cell Biology 2019). Not to even mention that 80 um/sec velocity for a vesicle has not been described in bibliography so far.

* We have noticed that serum deprivation reduces movement proportion and increases pauses, but accelerate velocities? This non expected increase in velocities should be further addressed and discussed.

* The number of neurons analyzed in each experiment is too small for axonal transport experiments with high variability and with not certain definitions of projections analyzed.

* In general, imaging methods are poor. Authors should state how pausing has been defined and how pausing time was calculated. In Figure 1 authors should state the n of particles analyzed and the number of pauses. Figure legends in general should be improved

* As it is observed from graphs, velocities are not distributed with a normal distribution. Have the authors used a test for normality? Why data is not analyzed as non-parametric data?

* Why do authors use HEK293T cells as control for some experiments and astrocytes for others? HEK293T cells are human derived, since primary cultures are from mice, this cell line is not the best control. The use of a mouse derived cell line or primary culture (like astrocytes) is a better option. Furthermore, it would be interesting to see how exosomes from primary astrocytes affect WT neurons in standard conditions as a control (in comparison to Oligodendrocyte exosomes).

* Despite CNP and PLP being mostly expressed in Shawn cells and oligodendrocytes (there is evidence of PLP expression in neurons (doi: 10.1002/jnr.22121)), it would be interesting to determine neuron mutant morphology and axonal transport and, if changes arise, the effect of WT EVs over these mutant neurons.

* It is not clear how mislocalization of MVB in mielinated tracks is correlated with abnormal exosome release by oligodencrocites since localization is differentially impaired between mutants, but both mutants showed reduced proteomic contents in O-EVs.

* Why do authors used a Student t tests instead of One Way ANOVA and a post-test for multiple comparisons?

* In discussion, authors state that the findings of this work proved that EVs have a long term effect on neurons, however experiments were not carried over periods longer than 24 hs.

* Since it has been proven that EVs modulate neurons metabolic rates, electrophysiology and now axonal transport, a discussion regarding axonal transport modulation being a direct effect of EVs or a secondary effect, subsequent to metabolic or electrophysiological changes should be included

Minor concerns:

* In introduction citations are missing. Furthermore, regarding to axonal transport.

* -What are red arrows pointing at?

* -Bar for distance and time frames are missing in figures.

* Are kymographs representative of the pictures showing time lapse progresion in the manuscript?

Rev. 4:

Kramer-Albers and coworkers previously showed that small EVs (exosomes) released by oligodendrocytes sustain neuron metabolism , especially under stress conditions. In this study they build on these previous findings and show that oligodendrocytes-derived sEVs promote axonal transport, a process that is essential for axonal integrity. In addition, they show that exosome production is quantitatively and qualitative impaired in oligodendrocytes from PrP- and CNP-ko mice, which exhibit secondary axonal degeneration.

Collectively, these novel data suggest that impaired exosome production from oligodendrocytes is responsible for alterations of axonal transport and consequent axonal degeneration in the mutant mice. By suggesting the mechanism underlying secondary axonal degeneration in PrP- and CNP-ko transgenic mice this excellent work has a high impact in the neurodegeneration field and provides a significant advancement in current understanding of how glial cells dysfunction contribute to axonal degeneration. In general the manuscript is well written, the experiments are well done and clearly described and appropriate controls are include

However, there are a few points that need to further explored or clarified

-While fig 1 shows the effects of short treatment with sEVs on axonal transport in control conditions, the effects of prolong sEV treatment (12-13 h) without H202 is not reported in fig 2. This makes impossible to verify whether or not Ols-derived sEVs completely or partially rescue the alterations of intra-axonal transport caused by stress and to appreciate the transport defects induced by H202. Does prolong treatment influence basal transport kinetics besides decreasing the pausing time? How prolong treatment influences vesicle transport (anterograde and retrograde transport and the percentage of static EVs….) in the absence of stressful stimuli should be included in fig 2. The same hold true for fig3 where the effect of treatment with sEV-conditioned medium in the absence of nutrient deprivation is undefined. In order to conclude that sEVs preferentially protect axons under stressful conditions, the action of sEVs /conditioned medium containing sEVs in control conditions should be shown.

- Why sEVs were resuspended in PBS before addition to neurons, given the cells were live imaged at 37°c and 5% CO2, thus likely in neuronal medium? Medium dilution with PBS may cause neuronal damage.

-It is not clear to me from the "sEV treatment and stress condition…" paragraph in the methods whether neurons were exposed to similar concentration of sEVs produced by HEK293T or astrocytes compared to oligodendrocytes. The number of particles produced by HEK293T and astrocytes should be directly estimated by NTA analysis.

-For clarity, define sEV-conditioned oligodendrocyte supernatant (10k) the first time it is mentioned in the results. To help the reader, make clear that the 10k supernatant is the sup after centrifugation of large EVs.

-Figure 4B-D, G, Fig 5 F Please report in the figs or in the legend whether changes in MVB distribution are statistically significant or not.

-Page 8, second line. Please clarify why the biological process "transport" fits with the biological activity of small EVs for readers not in the EV field

- Page 9. The authors refer to metabolic activity/cell viability of neurons without describing which functional parameter/assay they were measuring. Please make clear in the results that neuronal metabolism/integrity was evaluated by the MTT assay.

-With respect to neuron metabolism, the MTT assay is not enough to claim that EVs support the metabolic activity of neurons (as repetitively reported in the text, see for example the discussion at page 11) . Complementary approaches (e.g. Seahorse evaluation of oxygen consumption rate (OCR) and extracellular acidification rate) should be used to better define how sEVs influence neuronal metabolism. This would significantly strengthen the results of the study.

- Please clarify why cell viability was analyzed in cortical neurons while vesicle transport was explored in hippocampal cultures. It should be reported in the results, not only in the methods, that axonal transport was analyzed in immature, developing neurons. Moreover, the differentiation stage of neurons assayed by MTT has not been specified. Please add this info in to the text

---

## [Decision Letter · Decision Letter 2]

23 Sep 2020

Dear Dr Krämer-Albers,

Thank you very much for submitting a revised version of your manuscript "Exosomes derived from PLP- and CNP-deficient oligodendrocytes lack the ability to support neurons and fail to promote axonal transport" for consideration as a Research Article at PLOS Biology. Thank you also for your patience as we completed our editorial process, and please accept my apologies for the delay in providing you with our decision. This revised version of your manuscript has been evaluated by the PLOS Biology editors, the Academic Editor and three of the original reviewers.

The reviews are attached below. You will see that the reviewers find the manuscript much improved and that they think most of their concerns are now resolved. Nevertheless, Reviewer 2 raises some points that should be addressed either by the proposed immunobead exosome depletion experiment or by specifically stating in the discussion that the possibility that some of the effects observed are mediated by a factor co-purified with the exosomes cannot be categorically ruled out. Reviewer 3 would like you to clarify whether or not the same data are being presented more than once in Figures 2 and 6 – if this is the case, it should be fixed - and other points that can be addressed by improving the description of the methods and re-analyzing some of the data.

After discussing the reviews with the academic editor, we are pleased to offer you the opportunity to address the points raised by the reviewers in a revised version that we anticipate should not take you very long. We will then assess your revised manuscript and your response to the reviewers' comments.

We expect to receive your revised manuscript within 1 month.

**IMPORTANT - SUBMITTING YOUR REVISION**

*Resubmission Checklist*

*Published Peer Review*

*PLOS Data Policy*

*Blot and Gel Data Policy*

Sincerely,

Ines

--

Ines Alvarez-Garcia, PhD,

Senior Editor,

ialvarez-garcia@plos.org,

PLOS Biology

Reviewers’ comments

Rev. 2:

In this revised manuscript, the authors have made efforts to include new data to support the original conclusions. While the COVID pandemic has made it difficult to perform additional experiments to fully address my concerns, the lack of such experiments pose substantial ambiguity in whether it is sEV or other soluble factors that mediate oligodendroglial support to axons.

While the authors emphasize the comparison between 10k supernatant (+ exosome) and 100k supernatant ( -exosome), the ultracentrifugation is unable to separate sEV and other proteins that attach on sEV. The similar density distribution of sEV as described in previous studies does not support whether there are other proteins associated with the 100k pellet, as it depends on associated protein quantity. It would be imperative to test whether supernatant from immunobeads depleted exosomes still give functional phenotypes or not, the diminished effect from supernatant would add more convincing data to support the observed axon fast transport is indeed mediated by exosomes, but not associated other soluble factors.

In addition, the authors observed loss of functional mutant sEV data in axon transport. However, given the secondary axon degeneration phenotype in both mutant mice, the effect on reduced axon transport by mutant oligo-derived sEV remains insufficient to conclude that this would be the underlying mechanisms for axon degeneration. It should be demonstrated whether mutant oligodendrocytes derived sEV has potential direct toxicity to axons, which may directly link to the axon degeneration phenotype.

In my opinion, both issues are directly related to the main conclusion of the current manuscript. It is thus important to clarify these key questions. While the revised manuscript provided explanation on prior critiques, the lack of these experimental results makes it frail in interpreting key observations and undermines the novelty of the current study.

Rev. 3:

I appreciate the response and the efforts invested by the authors in answering reviewers' concerns and also understand the difficulties arising due to worldwide situation. I`m pleased to see that authors decided to adjusted title to findings, focused the work and made it more cohesive. I consider the work very relevant but would like to re-state some concerns that have not been properly addressed.

I consider that data duplication in figures should not be presented. It seems that what was previously shown as figure 7, it is now shown duplicated partially as Figure 2 and repeated again in Figure 6?.

It could be that I`m making a mistake in identifying this, and I`m sorry if what I`m saying is not correct. But in case it is, should not be presented this way.

I appreciate the clarifications included in methods for the identification of axons and cited seminal papers. As authors realized, axons at day 2 to 3 after neuronal plating are in their determination phase being only 1,5 longer than the longest dendrite. So, a statement is necessary to be included in text clarifying the early maturation stage of neurons.

I`m relief that velocities observed are now in literature range and would suggest to express them as um/sec to standardize it with most of the work on transport dynamics. It is still confusing whether authors are looking to segmental velocities or average velocities. And even, are plotted velocities correspond to average particle velocity or to kymograph average velocity?, since the procedure for this calculation is still not included in methods.

With respect to pauses, my advice is considering pauses to no movement within moving particles. As per your stated criteria a particle moving at lower than 0,5 microns per second is considered pausing but your average speed in control conditions is near 0,6 um/sec.

I`m pleased that result differences are maintained after performing statistical test for non parametric data when it is required. I would suggest that data without normality is presented as median, boxes and whiskers, instead of average and errors.

Rev. 4:

The authors satisfactorily addressed all the issues raised in my previous review.

---

## [Editor Report · Decision Letter 3]

21 Nov 2020

Dear Dr Krämer-Albers,

Thank you for submitting your revised Research Article entitled "Exosomes derived from PLP- and CNP-deficient oligodendrocytes lack the ability to support neurons and fail to promote axonal transport" for publication in PLOS Biology. I have now obtained advice from the Academic Editor and have discussed the comments with the team of editors.

We're delighted to let you know that we're now editorially satisfied with your manuscript. However, we would like you to include in the manuscript the clarification you recently sent us on the point regarding the replication data in Fig. 2 and 6. In addition, please consider our suggestion to improve the title and make it more accessible for a broad audience as follows:

"Oligodendrocyte genetic defects impair the secretion of glia exosomes and their ability to support nutrient-deprived neurons and promote axonal transport"

Before we can formally accept your paper and consider it "in press", we also need to ensure that your article conforms to our guidelines. A member of our team will be in touch shortly with a set of requests. As we can't proceed until these requirements are met, your swift response will help prevent delays to publication. Please also make sure to address the data and other policy-related requests noted at the end of this email.

- a cover letter that should detail your responses to any editorial requests, if applicable

*Copyediting*

*Published Peer Review History*

*Early Version*

Best wishes,

Ines

--

Ines Alvarez-Garcia, PhD,

Senior Editor,

ialvarez-garcia@plos.org,

PLOS Biology

Thank you for complying with our data policy and sending us the data underlying all individual quantitative observations summarized in the figures. However, we are still missing data from the following figures that you will need to provide:

Fig. 3E; Fig. 4A, B; Fig. S3C-D and Fig. S5

Please revise the Data Availability statements that you have included in the paper (“The data that support the findings of this study that are not included within the paper and its supplementary information files are available from the corresponding author upon reasonable request”), as it contradicts the statement reflected in the metadata section (“All relevant data are within the paper and its Supporting Information files”). All relevant data should be made available according to our policy.

---

## [Editor Report · Decision Letter 4]

10 Dec 2020

Dear Dr. Krämer-Albers,

I am writing concerning your manuscript submitted to PLOS Biology, entitled “Oligodendrocytes support axonal transport and maintenance via exosome secretion.”

We have now completed our final technical checks and have approved your submission for publication. You will shortly receive a letter of formal acceptance from the editor.

Kind regards,

PLOS Biology